# Recurrent introgression and geographical stratification shape *Saccharomyces cerevisiae* in the Neotropics

J. Abraham Avelar-Rivas [1,7], Iván Sedeño [2,3], Luis F. García-Ortega [4], Jose A. Urban Aragon [2,3,8], Claudio López-Gallegos [1,4], Xitlali Aguirre-Dugua [5], Eugenio Mancera [4,9] ✉, Alexander DeLuna [1,6,9] ✉ & Lucia Morales [2,9] ✉

From yeasts to humans, introgressive hybridization significantly influences the evolutionary history of living organisms by introducing new genetic diversity. Strains of *Saccharomyces cerevisiae* worldwide exhibit introgressions from the sister species *S. paradoxus*, despite the average sequence identity between these species being lower than 90%. While *S. cerevisiae* isolates from the Neotropics are known for their high levels of introgression, the hybridization events originating them remain unclear. Here, we sequence 216 *S. cerevisiae* isolates from open, spontaneous agave fermentation across Mexico. The genomes of these strains reveal considerable genetic diversity and population structure linked to geographic distribution, which had been overlooked due to undersampling of this megadiverse region. These strains, along with those from French Guiana, Ecuador, and Brazil, form a broader Neotropical phylogenetic cluster that is notably enriched in introgressions. Surprisingly, their origins and the observed conservation patterns of these introgressions indicate multiple hybridization events, suggesting flexible species barriers in this region. Our findings underscore concurrent evolutionary processes—geographical stratification and multiple introgressions—that shape the genomes of a diverse lineage of *S. cerevisiae*. Neotropical yeasts thus provide a natural laboratory for exploring the mechanisms and adaptive significance of introgressive hybridization in eukaryotic genome evolution.

Beyond its long-established status as the most extensively studied eukaryotic-cell model in the laboratory, the budding yeast *Saccharomyces cerevisiae* (*S. cerevisiae*) has emerged as an outstanding model for studies in ecology, population genomics, and evolution[1–5]. Genomic analyses of thousands of isolates from natural and human-related environments around the world reveal that the species is structured in clades that broadly correlate with geography and isolation source[6–8]. These global genomic studies have also shown that most strains harbor genetic material from other *Saccharomyces* species, a

phenomenon referred to as genetic introgression. For instance, most *S. cerevisiae* isolates from around the world harbor genes introgressed from its sister species *Saccharomyces paradoxus* (*S. paradoxus*)[6,9–12], even though these species diverged 4–6 MYA and have an average nucleotide divergence of ~14%[13,14].

The neotropical region, which extends from northern Mexico to South America, is biogeographically relevant due to its rich biodiversity[15]. Some of the *S. cerevisiae* strains from this area group in a long-monophyletic branch referred to as a complex clade by Pontes et

al.[7]. This branch aggregates four groups of strains: Mexican agave (MA), French Guiana (FG), Wild Brazil 3 (WB3) and South American mix 2 (SAM2). While WB3 strains were isolated from natural areas, most of the strains from this Neotropical cluster were isolated from anthropogenic environments. Considering the vast area encompassed by this region, the few available sequenced isolates likely provide an underestimate of genetic diversity, thus limiting our understanding of the species in this part of the world.

The handful of sequenced genomes available indicated that the *S. cerevisiae* Neotropical cluster harbors an unusually high number of introgressions from *S. paradoxus*[6,7,12,16,17]. This could explain the high divergence of the cluster when compared to other clades. A fraction of the introgressions from the Neotropical cluster has been associated with populations of *S. paradoxus* from the Americas[6,12,16], which contrast with the origin of the introgressions in the European linages[6,18]. It has even been suggested that the introgressions in Brazilian strains derive from two different *S. paradoxus* populations, raising the possibility of multiple hybridization events[16]. However, the precise origins of introgressions remained unclear in part due to the limited number of sequenced *S. cerevisiae* and *S. paradoxus* genomes available from the region. In Mexico, there are reports of *S. paradoxus* consistently at low frequency in agave fermentations[19–23], including a lineage that coexists with *S. cerevisiae* in the agave distillery environment and that has not been considered by previous studies[23]. This lineage could be a potential donor for introgressions in the tropical Americas. Comprehensive understanding of both the vertical evolution of *S. cerevisiae* and the origins of its introgressions is essential to accurately determine the dynamics of interspecies gene flow in the region.

To explore the genomic diversity of *S. cerevisiae* in the tropical Americas—particularly in light of its strong signatures of introgression—we sequence 216 strains primarily isolated from traditional agave fermentations in Mexico. These spontaneous fermentations are anthropogenic, open systems that rely on microorganisms from the surrounding, often natural, environments[22,23]. Previous studies have highlighted the megadiversity in this region[24] as well as the high diversity of *S. cerevisiae* in agave fermentation[25–28], but whole-genome sequencing efforts have only included nine strains, mostly from a single agave fermenting region[6]. We show that most of the newly sequenced isolates belong to the Neotropical cluster, revealing new clades where diversity correlates with the geographic origin of the strains. Moreover, by analyzing the patterns and origins of the introgressions in the context of the relatedness of these lineages, we identify multiple introgression events in their evolutionary history. Notably, some of these introgression pulses are unique to strains isolated from agave fermentations. Our refined phylogeny allows us to illustrate how recurrent episodes of gene flow between two sister *Saccharomyces* species have unfolded in a megadiverse region of the world.

## Results

To provide a comprehensive view of *S. cerevisiae* diversity in one of the most biodiverse regions in the world[15,24], we sequenced the genomes of 216 isolates primarily isolated from traditional agave distilleries across Mexico (Supplementary Fig. 1 and Supplementary Data 1). This anthropogenic fermentation environment provides a good starting point to study yeast diversity, as spontaneous fermentations have taken place for over 3500 years in a wide variety of ecosystems, latitudes, and cultural contexts[22,29,30]. Specifically, 211 of the sequenced strains were isolated from agave fermentations collected through various sampling efforts conducted across the country between the years 1988 and 2021[20,23,25,26,28,31–33]. Four of the five remaining isolates were from traditional fermentations of other substrates—pulque (fermented raw agave sap), apple cider, cactus pears, and cooked stems of sotol plants (*Dasylirion* spp.)—while the fifth was an uncharacterized commercial yeast strain used in one of the sampled distilleries; all of

them were isolated in Mexico (Supplementary Data 1). With the 216 genomes (median depth coverage = 163X), we assessed the phylogenetic position and population structure of the strains in the context of the known diversity of the species, and we also analyzed the dynamics and origins of the prevalent *S. paradoxus* introgressions within these populations.

### Agave fermentation strains belong to a genetically diverse Neotropical cluster

To assess the relatedness of the agave *S. cerevisiae* strains with other isolates of the species, we first performed multidimensional scaling analysis (MDS) with 1262 genomes, including the 216 newly sequenced strains. Most isolates from Mexico clustered next to other strains from the tropical Americas, such as those from the FG, the WB3, the SAM2, and the Mexican agave 1 (MA1) (formerly named "MA") clades (Fig. 1a). The majority of the strains isolated from the tropical Americas form a consolidated group, separated from other clades of the world (Fig. 1a), a pattern that persisted even when strains from agave fermentation are not overrepresented in the MDS analysis (Supplementary Fig. 2). Only eleven of the sequenced strains here reported grouped with clades outside the Neotropical cluster. Meanwhile, all four isolates that we sequenced from non-agave substrates clustered with the agave-fermentation strains from the Neotropics.

To establish the phylogenetic relationships of the newly sequenced isolates, we compared them to a reference panel of strains (see Supplementary Data 2). Consistent with the MDS analysis, the phylogeny showed that 205 of the 216 newly sequenced strains (94.9%) were part of a monophyletic group that included two newly described clades, Mexican Agave 2 (MA2) and Tequila Distillery, along with the previously defined clades MA1, FG, WB3, and SAM2 (Supplementary Fig. 2). The genetic differentiation of MA1, MA2, WB3, and FG was supported by weighted $F_{st}$ analyses (Supplementary Fig. 2). We refer to this higher hierarchy clade as the Neotropical cluster of *S. cerevisiae*, as nearly all the strains in this group come from the tropical Americas (Fig. 1b). Among the remaining sequenced isolates, five grouped with the North American Oak clade, which are commonly classified as wild strains[6,8,34]. Another five isolates—including the commercial strain—were closely related to the Mixed Origin clade and only one clustered with Wine strains.

The Neotropical cluster consistently exhibited a relatively long branch in the phylogeny, encompassing all strains from six clades: MA1, MA2, FG, WB3, Tequila Distillery, and SAM2. The long branch was supported by the high number of single nucleotide variants (SNVs) present in the strains of these clades (Fig. 2). In some of the phylogenetic reconstructions, the inclusion of the mosaic strains of Tequila Distillery and SAM2 changed the relative placement of MA1, MA2, WB3, and FG (Supplementary Fig. 2). These and other mosaic strains were excluded from the main phylogeny (Fig. 1b) to avoid artefactual topological shifts driven by admixture. To determine whether the extended branch lengths observed in the Neotropical cluster were driven by introgressed regions, we generated the phylogeny excluding introgressions (Methods). Removal of these regions had no impact on the overall topology of the phylogeny and exhibited only a modest effect on the branch lengths of the Neotropical cluster (Supplementary Fig. 2). The number of SNVs of the neotropical groups slightly diminished when introgressed regions were removed (Fig. 2). In contrast, removal of introgression markedly reduced branch lengths and SNV counts in the Alpechin clade, a group of European strains that harbors a high number of *S. paradoxus* introgressions. To assess whether the observed long branches were attributable to sampling bias of strains, we reconstructed the phylogenies using random downsampling to a maximum of five strains per clade (Supplementary Fig. 2). Resulting phylogenetic topologies consistently showed the same position and relatively long branches of the Neotropical cluster, although the more balanced dataset did slightly reduce the relative branch length

 

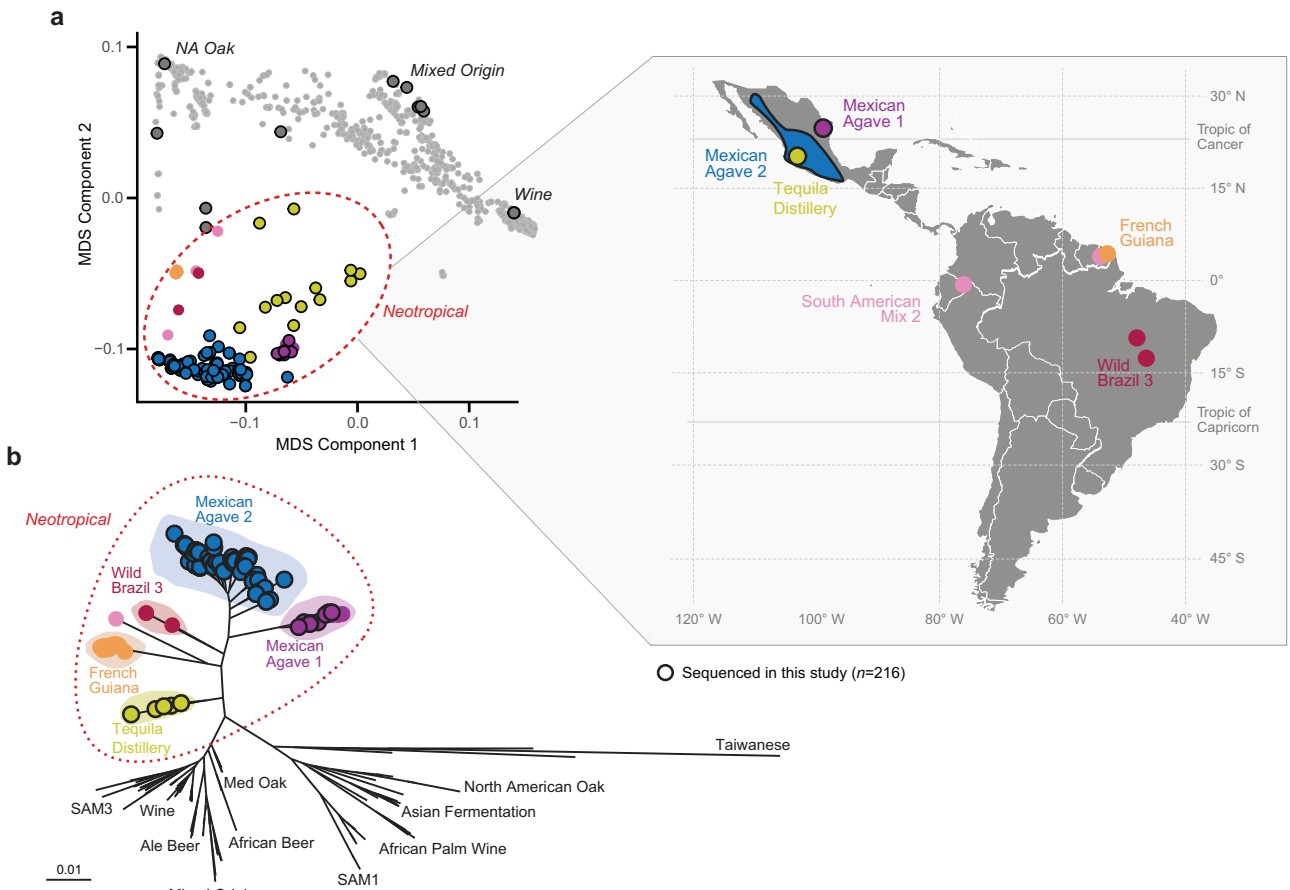

**Fig. 1 | A novel, divergent yeast lineage encompasses most of the diversity in open spontaneous agave fermentations. a** MDS analysis of 1262 genomes from strains around the world based on 173,331 informative sites. Large dots indicate strains from the tropical Americas, colored according to their clade: "Mexican Agave 1" (purple), "Mexican Agave 2" (blue), "French Guiana" (orange), "Tequila Distillery" (yellow), "Wild Brazil 3" (carmine) and "South American Mix 2" (SAM2, pink). Small gray dots indicate isolates from other parts of the world, with the Wine, Mixed Origin and North American Oak (NA Oak) shown as references. Black solid outlines are strains sequenced in this study (*n* = 216). Gray circles with a solid outline indicate the few isolates from agave fermentation that did not group with other neotropical strains. **b** Maximum-likelihood phylogenetic tree inferred using 694,264 SNPs across 332 strains, including a subset of representative reference genomes from strains from around the world. Admixed strains identified by ADMIXTURE analysis were excluded (Fig. 3a, see Methods). For clarity purposes, only key clades are labeled, although all relevant lineages of the species were included in the phylogeny (see Supplementary Data 2). Clade names are according to Tellini N. et al.[12]; for the original names of other neotropical clades, see Barbosa et al.[16], Barbosa et al.[40], and Peter et al.[6]. Source data are provided as a Source data file.

(Supplementary Fig. 2), consistent with a previous report[34]. Collectively, these results highlight the robustness of the clades within the Neotropical cluster and its characteristically long branch, consistent with previously reported global phylogenies of *S. cerevisiae*[6,8].

As mentioned above, three clades could be distinguished among the newly sequenced isolates from agave fermentations in neotropical *S. cerevisiae*. First, we identified a group of 20 isolates from northeastern Mexico, which included the seven strains previously reported as "Mexican Agave"[6], herein referred to as the MA1 clade (Fig. 1b). It is worth mentioning that the monophyletic MA1 clade comprises so far only strains isolated northeast of the Sierra Madre Oriental, a mountain range which runs parallel to the coast of the Gulf of Mexico and acts as a natural barrier, exerting strong influence on the climate and biodiversity on either side of the range[35,36]. As a result of the comprehensive sampling of the isolates reported here, we identified a novel MA2 clade that is closely related to MA1 (Fig. 1b). Most of the newly sequenced strains in this study (81.9%) belonged to MA2 clade. Two previously sequenced strains isolated in Mexico[6] formerly assigned to the SAM2 clade[12], were reclassified after these analyses within the new MA2 clade. The third group of agave isolates was composed of 15 strains, most of them with admixed genotypes, all sequenced in this study (Fig. 1b and Supplementary Fig. 2). We refer to this group as "Tequila Distillery", since eleven of the 15 strains were isolated from the tequila industry.

Our results show that most strains from spontaneous agave fermentations are part of the Neotropical cluster, revealing that the group previously described as "Mexican Agave" represents a fraction of the genetic diversity present in agave fermentations. Furthermore, the clades within this cluster, including the newly identified groups, rank among the highest in terms of SNVs when compared to the reference genome, surpassed only by the wild Asian clades (Fig. 2 and Supplementary Fig. 3). This trend persisted when pairwise identity by sequence was quantified (Supplementary Fig. 3), indicating that it is independent of the specific reference genome used. The high genetic divergence of the neotropical strains was recently supported by a pairwise comparison of 3334 *S. cerevisiae* genomes, which revealed that the greatest pairwise genetic distance within the species occurs between a wild Taiwanese isolate and a Mexican strain from agave[8]. Together, our phylogenomic and population analyses of *S. cerevisiae* from the tropical Americas bridge the gap of genetic knowledge for the species in this region.

### Neotropical yeast populations are stratified by geography
Yeasts from the tropical Americas come from a diverse variety of environments across a broad range of geographical distances. We thus asked to what extent geographic parameters influence the population structure of *S. cerevisiae* in the Neotropical cluster. To this end, we

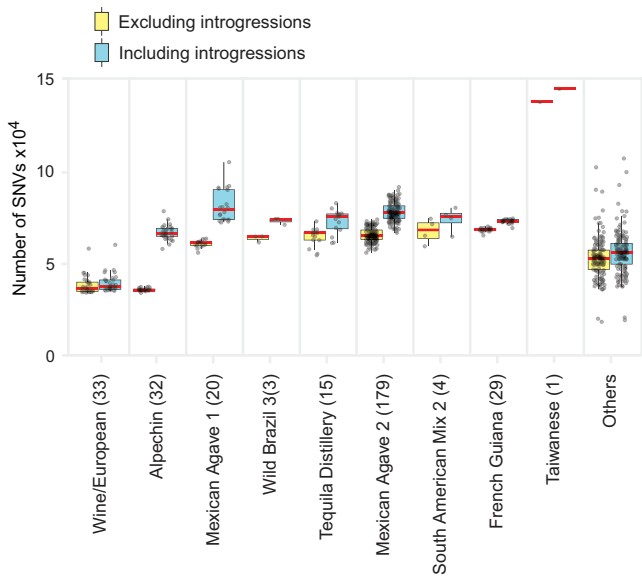

**Fig. 2 | Divergence of the Neotropical cluster is not only due to introgressions.** Box plots show the 25th and 75th percentiles, with whiskers extending to the minimum and maximum values within 1.5× the interquartile range. Points represent individual strain observations. Red horizontal lines indicate the medians of SNV counts for each phylogenetic clade; the number of strains per clade (*n*) is shown in parentheses (487 strains total). SNVs were called either from alignments to the *S. cerevisiae* reference genome (blue; including introgressed regions) or to the *S. cerevisiae* subgenome using a concatenated multispecies reference (yellow; excluding introgressed regions) (see Methods). Source data are provided as a Source data file.

carried out an ADMIXTURE analysis, selecting the number of populations that best explained the genotypes based on the lowest cross-validation error (Fig. 3a and Supplementary Fig. 4). The population structure analysis revealed eleven ancestral components within the Neotropical cluster. Specifically, strains from the MA1, Tequila Distillery, FG, and WB3 clades each formed distinct populations, while the remaining seven populations were all from the newly identified MA2 phylogenetic clade.

Given the stratification observed in the Neotropical cluster, we analyzed the geographical distribution of these isolates according to their population structure. The eleven colors in the ADMIXTURE plot representing ancestral components within the neotropical cluster showed good agreement with their geographic distribution (Fig. 3a). This geographic stratification was observed at two levels of resolution. First, each of the five clades of the Neotropical cluster are constrained to specific regions in the map. Second, at the resolution within the MA2 clade, the seven colors in the ADMIXTURE were also stratified by geography. In general, strains from the same agave-spirit producing region grouped together in the phylogeny and had the same ancestral population component. This was also demonstrated by a correlation between geographic and genetic distances in this set of strains ($r = 0.18$, significance 0.0001 with 9999 permutations, Spearman Mantel test, one-sided, including MA1 and MA2 strains). We also observed increasing genetic diversity and heterozygosity from north to south within the MA2 clade (Supplementary Fig. 5). Together, these findings suggest that the genomic diversity in the Neotropical cluster is influenced by geography.

Despite the overall agreement between geography and population structure in the strains from the tropical Americas, there were exceptions suggesting gene flow within and even beyond the Neotropical lineage. For instance, a strain isolated from a spontaneous agave fermentation shared close ancestry with wild isolates of the North American Oak clade, while other four isolates closely related to

the North American Oak clade showed signs of admixture with different genetic components of the neotropical cluster (Fig. 3a). The WB3 showed little admixture, while the SAM2 strains had genetic components of MA2, FG, and WB3 populations, which is not surprising since they are close relatives. Moreover, ten strains from the Tequila Distillery group showed admixture, mostly between different neotropical populations, but also with other clades, such as Wine. Taken together, these findings suggest that *S. cerevisiae* populations in the tropical Americas have diverged, at least partially, through geographic isolation, while other ecological processes or human-related factors have promoted gene flow, also shaping the genetic structure of the population.

We used the genomic data to further explore the patterns of genetic diversity in the neotropical strains. Each clade differed significantly in nucleotide diversity (π), Tajima's D, and heterozygosity (Fig. 3b–d and Supplementary Fig. 6), with some exhibiting high genetic diversity despite most of their strains being isolated from anthropogenic environments. For instance, the MA1, MA2, and Tequila Distillery clades all exhibit higher nucleotide diversity than the Wine clade and even than the highly introgressed Alpechin clade (Supplementary Fig. 6). These findings suggest that the elevated genetic diversity observed in the agave-related clades cannot be attributed to introgressions alone. In contrast, the FG clade showed low genetic diversity and a homogenous degree of heterozygosity. The Tequila clade displayed the highest levels of heterozygosity and nucleotide diversity, partly due to the admixed genotypes of its strains, and consistent with trends observed in domesticated beer isolates[6]. Taken together, our population-structure and genetic-diversity analyses indicate that yeasts in the tropical Americas constitute diverse, structured populations stratified by geography.

## Differences in the distribution of introgressions reveal recurrent interspecies gene flow

The genomes of *S. cerevisiae* strains within the Neotropical cluster are characterized by an atypically high number of introgressed regions from *S. paradoxus*[12,16]. To gain insight into the dynamics of gene flow, we identified the set of introgressed genes with a strategy that uses competitive mapping of sequencing reads against both parental genomes and phylogenetic analyses (Methods). Importantly, most of the introgressions that we identified were also found using other strategies, including: (i) genome assembly followed by a phylogenetic approach[6], (ii) calling bona fide single-nucleotide diagnostic markers of introgressions[12] or (iii) filtering orthogroups after competitive mapping[16] (Supplementary Fig. 7).

All Neotropical clades showed more introgressed genes than the other groups worldwide, except for the Alpechin clade (Fig. 4a). The three clades from South America (WB3, SAM2, and FG) displayed fewer and shorter introgressions with a lower fraction of heterozygous introgressed genes, compared to the three agave clades (Tequila Distillery, MA1, and MA2) (Fig. 4a and Supplementary Fig. 8). Here, we call an introgressed gene heterozygous when both *S. cerevisiae* and *S. paradoxus* alleles are present within the same genome (Fig. 4a; Supplementary Figs. 8 and 9).

Among the Neotropical groups, we found that MA1 isolates exhibited both the highest number and the widest range in the number of introgressed genes, with counts going from 94 to 320 per strain. MA2 isolates featured between 64 and 122 introgressed genes (Fig. 4a). Moreover, we identified a subset of strains from the MA1 clade with more than 150 introgressed genes exhibiting longer and more heterozygous introgressions than most strains of the clade (Fig. 4 and Supplementary Fig. 8).

To infer the history of introgressed genes, we analyzed their retention patterns both within and between clades. We observed that the presence-absence patterns clustered the three agave associated clades together and clearly separated them from a second cluster

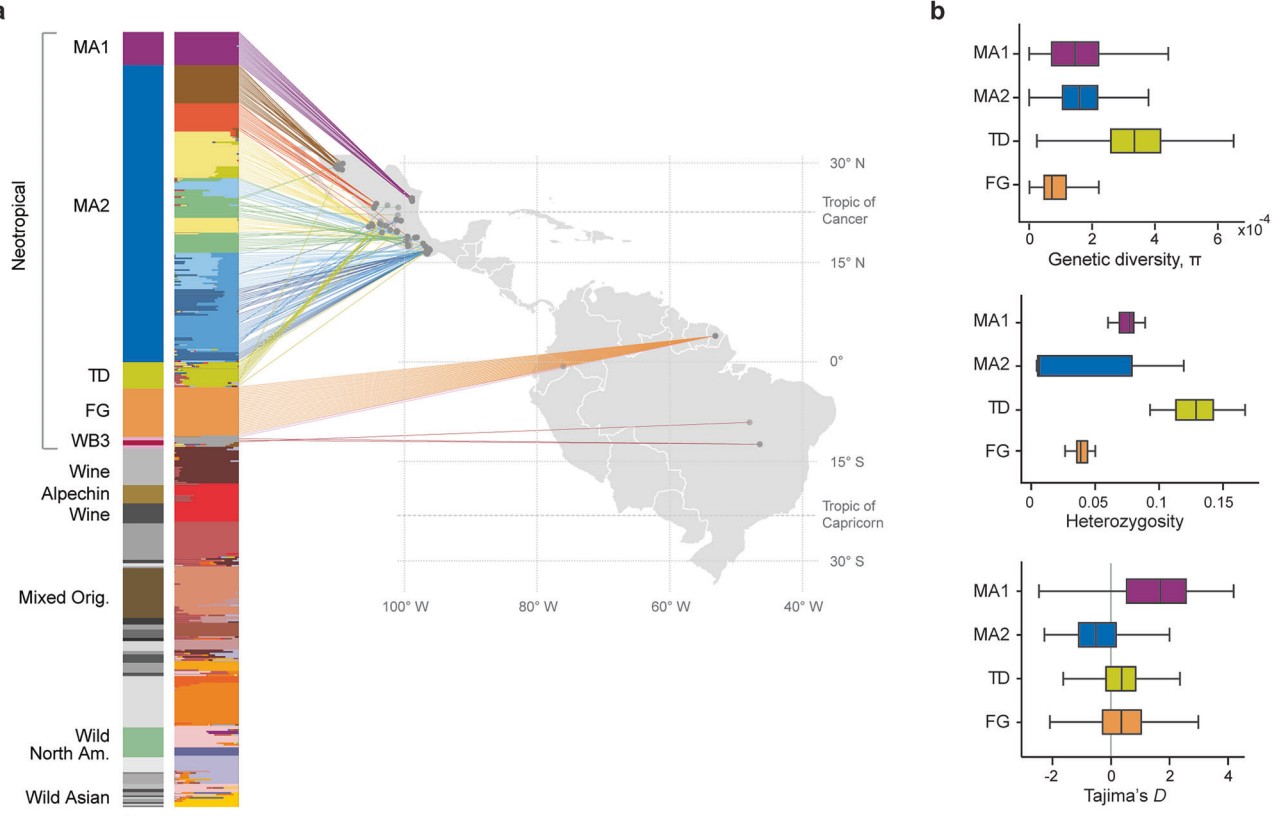

**Fig. 3 | Strains from the Neotropics show population structure correlated with geography. a** Population structure of strains from the Neotropics and other regions of the world as reference at $K = 24$ made with ADMIXTURE ($n = 466$); $K = 24$ was the $K$ with the lowest cross-validation error (Supplementary Fig. 5); strains are sorted according to phylogenetic clades (first track from the left). To the right, the geographic locations of the strains are shown on a map of the tropical Americas, with sampling sites in Mexico, French Guiana, Ecuador and Brazil. The colors of the lines connecting strains to their geographic locations correspond to the most prevalent genetic component of the strain in the ADMIXTURE plot (second track from the left). MA1 Mexican agave 1, MA2 Mexican agave 2, TD Tequila Distillery, FG French Guiana, WB3 Wild Brazil 3. **b** Genetic diversity (π), **c** heterozygosity and **d** Tajima's D of clades within the Neotropical cluster. Box plots show the 25th and 75th percentiles, with vertical lines indicating the median values. Whiskers extend to the minimum and maximum values within 1.5× the interquartile range; more extreme values are not shown. The number of strains considered for each clade are $n = 20$ for MA1; $n = 178$ for MA2; $n = 15$ for TD; $n = 29$ for FG; only the clades with more than 10 available sequences are shown. For the genetic diversity and Tajima's D, the number of 10-kb windows used for the box plots are MA1 $n = 1135$, MA2 $n = 1145$, TD = 1144, and FG $n = 1139$. Source data are provided as a Source data file.

comprising the three South American groups (SAM2, WB3, and FG. Fig. 4b; Supplementary Data 3 and Supplementary Fig. 9). Despite the differences between these two clusters, we identified significant overlap among most of the comparisons between subsets of the Neotropical clades (Supplementary Data 3), including five genes that are common to all six Neotropical clades. This number of shared introgressed genes exceeded the number expected by chance (hypergeometric test, one-sided, $p = 0.00234$; Supplementary Data 3). This is likely explained by an introgression pulse in the common ancestor of all Neotropical strains although strong selection could also account for the presence of this shared set of introgressed genes. Yet, no significant GO enrichment was detected among the genes present in at least 75% of strains in each clade. As expected, analysis of the presence–absence patterns of introgressed genes in the Alpechin clade, used as a reference, revealed no significant overlap with the neotropical groups, confirming the previously reported independent hybridization event underlying this clade[12,18].

Given the complex introgression patterns observed in the MA1 clade—multimodal distribution of number of introgressed genes (Fig. 4a), the presence of large heterozygous introgressed segments (Supplementary Fig. 8) and an unusually high number of genes exclusive to this clade but present at low frequency within it (Fig. 4b)— we examined the retention patterns alongside the phylogenetic relationships of the MA1 strains (Supplementary Fig. 9). While most strains

shared the majority of their introgressed regions with closely related isolates, seven MA1 strains exhibited heterozygous introgressed segments that were exclusively found within restricted phylogenetic groups. This pattern could be explained by multiple convergent losses of the same introgressed genes in other branches of MA1 and in MA2, or more likely, by at least one additional interspecies gene flow event from *S. paradoxus* that occurred specifically in a subset of MA1 strains.

### Introgressions in the Neotropical cluster trace to different lineages of *S. paradoxus*

To trace the origins of the observed introgressed regions, we inferred the most likely *S. paradoxus* lineage associated with each gene. To do so, we assessed the identity by sequence of each introgressed block against a panel of *S. paradoxus* genomes with representative strains from the Americas. This panel includes a diverse set of strains representing the previously defined populations from North America (*SpA*, *SpB*, *SpC*, and *SpD*), as well as a Brazilian *SpB* strain (*SpB_Bra*), along with representative strains from other established clades from China, Far East Asia, Hawaii and the reference strain from the European lineage (here grouped with *SpA*). Moreover, we included all sequenced strains from Mexico, encompassing three lineages exclusive to this region: *SpB_MxAgave*, *SpB_Mx1* and *SpB_Mx2*[23]. Our results indicate that most introgressed genes in the MA clades originated from *SpB_MxAgave* (Fig. 5a), a population found exclusively in agave distilleries in

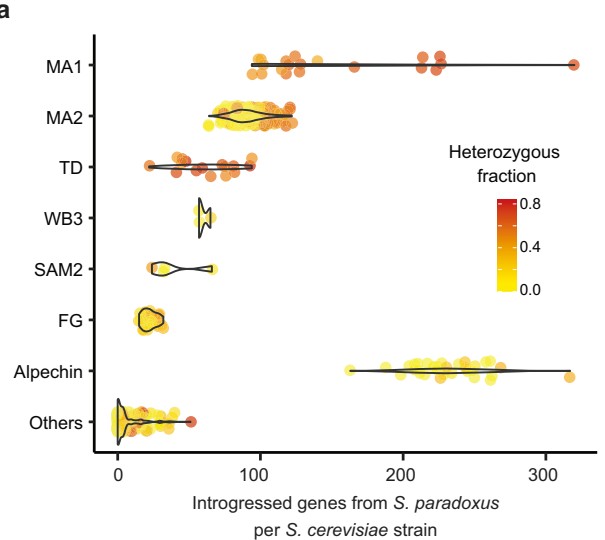

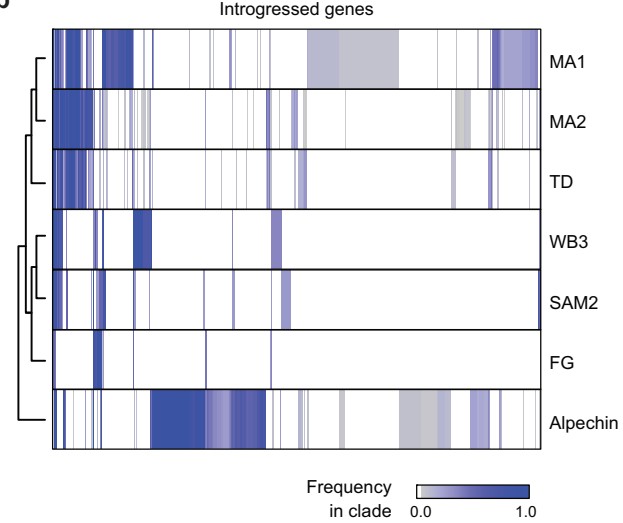

**Fig. 4 | Different patterns of introgressions in the clades from the Neotropics.**
**a** Distribution of the number of introgressed genes per strain in each group. Alpechin strains are included as a reference representing an independent, highly introgressed population. Dot colors (see color scale) indicates the fraction of introgressed genes that are heterozygous in each strain, defined by the presence of both *S. cerevisiae* and *S. paradoxus* alleles. **b** Clustering of introgressed genes identified across the Neotropical cluster and the alpechin clade (1226 genes in total, including 832 in Neotropical groups). Colors indicate the fraction of strains within each clade carrying the introgressed gene (see color scale). MA1 Mexican agave 1, MA2 Mexican agave 2, TD Tequila Distillery, WB3 Wild Brazil 3, SAM2, South-American mix 2, FG French Guiana. Source data are provided as a Source data file.

Mexico. A smaller fraction of introgressed genes in these strains appears to derive from the *SpB_Mx2* and the broader *SpB* lineage. In contrast, the majority of introgressed genes in the South American strains were traced to the *SpB* lineage, which is commonly found in natural environments in the Neotropics but has not been reported in agave fermentations[23]. As expected, introgressed genes in the Alpechin group most closely resembled those from the European lineage, previously identified as the donor population in this clade (Supplementary Data 4)[12,18,37]. These findings indicate that at least two lineages of *S. paradoxus* have hybridized with ancestors of the extant neotropical strains, highlighting the role of interspecies interactions in shaping the evolutionary trajectory of *S. cerevisiae* in the region.

## Discussion

Comprehensive sequencing of *S. cerevisiae* genomes in megadiverse, under-sampled regions is essential to understand the genomic diversity of this species, its natural evolutionary history and its association with domestication. In this study, we sequenced 216 strains from open agave fermentations, generating an extensive population genomics dataset of isolates from anthropogenic environments in the Neotropics. Our phylogenetic and population structure analyses reveal multiple interspecies gene-flow events resulting in introgressions within this geographically structured complex clade.

Introgressions impact evolutionary trajectories across the tree of life, from bacteria to humans[38,39], and are central to the evolutionary history of *S. cerevisiae*[6,10–12]. In the Americas, several *S. cerevisiae* lineages feature high number of introgressions from *S. paradoxus*, both in wild and anthropogenic environments[7,12,16,40,41]. Here, we provide a broad picture of the evolutionary dynamics of introgression happening in the tropical Americas with at least three introgression pulses. An early introgression likely occurred in the common ancestor of the Neotropical cluster before its diversification, where genes of the American *SpB* lineage of *S. paradoxus* introgressed into *S. cerevisiae* (Fig. 5b). This first pulse was followed by a subsequent introgression event from another lineage, *SpB_Mx*, in the common ancestor of MA1 and MA2. The presence of large heterozygous introgressed regions exclusive to a subset of MA1 strains suggests an additional, more recent pulse of introgression exclusive to this clade. Whether these

multiple introgression pulses are stochastic outcomes of the coexistence of *S. cerevisiae* and *S. paradoxus* in the region, or reflect adaptative processes, remains unclear, particularly given that we found no enrichment for specific functional categories among the commonly retained introgressed genes within clades.

The number and identity of introgressed genes vary widely between MA1 strains. This is consistent with an ongoing transition from hybridization to introgression, likely shaped by genome instability and backcrossing[18]. Variation in introgressed-gene content may result from differential allelic retention during genomic instability, different number of backcrossing events and outbreeding between strains at different stages of the transition from hybridization to stable introgressed genes. Overall, the variable patterns of introgressed genes in MA1 are difficult to reconcile with differential retention from the hybridization event shared with MA2 or fewer backcrossing events, though this cannot be ruled out entirely. Experimental and computational tests may help estimate the timing and the relative contribution of these scenarios, while broader sampling in this region would also likely provide deeper insights into the evolutionary processes shaping interspecies gene-flow in *Saccharomyces*.

Our sequencing effort expanded knowledge of *S. cerevisiae* diversity and biogeography. We identified two additional clades within the Neotropical cluster: MA2 and the admixed Tequila Distillery, which together with MA1, FG, WB3 and SAM2, comprise six distinct neotropical groups. Phylogenetic analysis indicated that the two MA clades−MA1 and MA2−are sister clades, which in turn are the closest relatives of the WB3 and FG strains, in agreement with the previously reported phylogeny[42]. The marked heterogeneity observed among the few isolates from the undersampled neotropical groups suggests the existence of additional, yet undiscovered lineages across the tropical Americas, a region of exceptional diversity and biogeographical complexity[15]. Expanded environmental sampling and genomic analysis in this region will be essential to uncover the full extent of *S. cerevisiae* diversity and the evolutionary forces shaping its natural history.

We showed that the introgressed segments increase branch lengths in the phylogeny, but the branch length of the Neotropical cluster was less affected by the removal of introgressions than that of the Alpechin group. Although branch length can also be influenced by

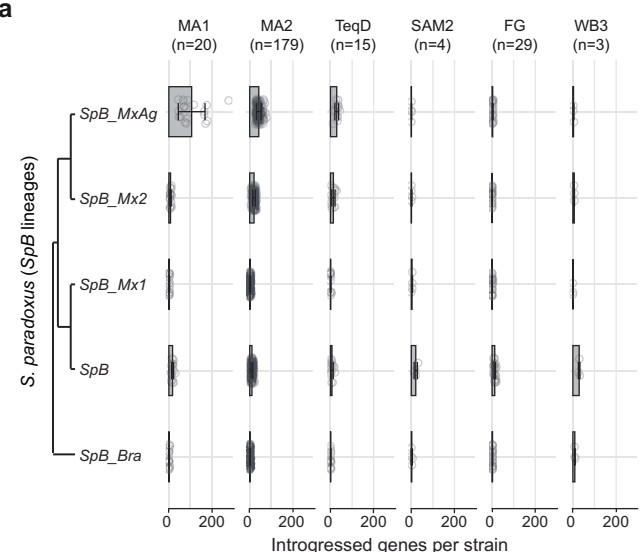

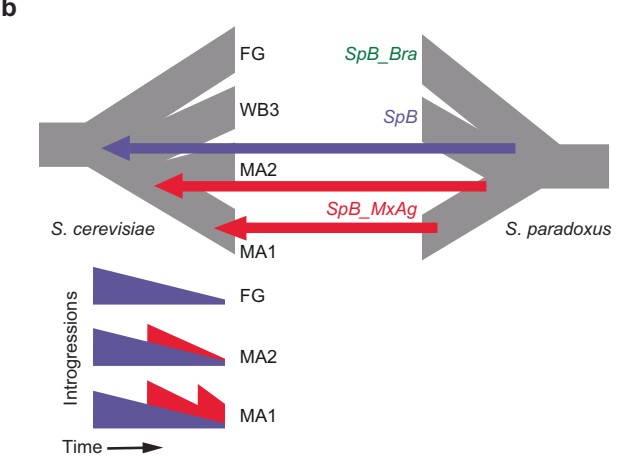

**Fig. 5 | Multiple introgression pulses from two *S. paradoxus* lineages in the neotropical *S. cerevisiae* cluster. a** Number of introgressed genes from each lineage of *S. paradoxus* in Neotropical *S. cerevisiae* clades. Each open circle represents an individual genome; gray bar indicates the mean, and the error bars represent one standard deviation. **b** Schematic model illustrating multiple introgression pulses (purple and red arrows) as the most plausible evolutionary scenario to explain the observed patterns in the number, heterozygosity, length, and *S. paradoxus* origin of introgressed blocks. MA1 Mexican agave 1, MA2 Mexican agave 2, TD Tequila Distillery, SAM2 South-American mix 2, FG French Guiana, WB3 Wild Brazil 3. Source data are provided as a Source data file.

the samples considered in the phylogeny, the Neotropical cluster remained among the most divergent lineages when we down sampled to have the same number of strains per clade (Supplementary Fig. 2). The extended branch of the Neotropical cluster may stem from multiple factors, including an elevated mutation rate caused by a hyper-mutator phenotype[43] or mutagen exposure, as well as demographic processes, such as changes in effective population size or increased outcrossing rates. Importantly, we found that the fraction of the genetic diversity in the Neotropical cluster that was previously unknown is structured and associated with the geographic origins of the yeast isolates. Geographical influence that shaped the genetic diversity was evident across several dimensions: First, a clear correlation was observed between each of the eleven ancestral components of the strains from the Neotropics and their region of isolation.

Second, strains from the MA2 clade exhibit increased nucleotide diversity and heterozygosity in a north-to-south gradient. Finally, the genetic divergence between the MA1 and MA2 clades ($F_{st} = 0.43$, median IBS = 0.97) is associated with the presence of the Sierra Madre Oriental mountain range, which is found at the edge where the neotropical and nearctic biogeographic regions coincide. This range is known to act as a geographical barrier impacting the distributions of multiple species[15,35,36].

Traditional agave fermentation systems are anthropogenic environments where fermentation takes place spontaneously in open tanks[22]. The *S. cerevisiae* sequenced strains from agave fermentations have been classified as domesticated based in their isolation source and on genetic signatures[7,17]. However, the boundaries between fermenting microbial communities and those from surrounding environments are often blurred in these and other open systems. For example, *S. cerevisiae* isolates collected from natural environments adjacent to distilleries are genetically similar to those from spontaneous open agave fermentations within the same area[23]. Additionally, some isolates from agave fermentation clustered with North American Oak strains, a clade typically classified as wild[6,8,34]. Moreover, certain subpopulations from fermentations within the MA2 clade were predominantly homozygous, which is a genetic signature previously associated with wild populations. Together, these findings suggest that the boundaries between *S. cerevisiae* from agave fermentations and those from the surrounding natural habitats are weak.

Beyond biogeographical factors, human activities may have played a role in shaping these yeast populations, as evidenced by admixture events within and beyond the Neotropical cluster. It remains to be addressed to what extent the observed population structure is driven by natural factors, such as the ecology and biogeography of plant reservoirs and insect vectors or by human dynamics and practices. For instance, specific production practices at the local level and the remarkable diversity of fermentation substrates, with over 50 different *Agave* species used for producing spirits[44], may be driving population differentiation. The composition of microbial communities in agave fermentations has previously been shown to be shaped by local factors[45]. With the rising global demand for agave spirits, there is a growing risk that producers may abandon traditional fermentation practices. This shift could lead to the loss of the microbial diversity herein described. This work provides a critical framework for developing conservation and management strategies for these fungi, highlighting their value beyond basic research. The agave fermentation system is not only a reservoir of remarkable yeast genetic diversity, but also a unique natural laboratory to gain insights into the interplay of genetic, ecological and evolutionary factors shaping population differentiation, and interspecies introgression.

## Methods

### Genome sequencing and variant calling
DNA was purified with the MasterPure DNA purification kit as recommended by the manufacturer, and was sequenced using DNBSeq (2 × 150 bp; BGI, China). Raw reads were quality assessed with fastp V0.20.0[46]. For each isolate, the filtered reads were aligned with BWA-MEM v0.7.4[47] to a concatenated species reference genome that includes *S. cerevisiae* S288C, *S. paradoxus* YPS138, *S. mikatae* IFO1815, *S. kudriavzevii* Cr85, *S. jurei* M1, *S. arboricola* H6, *S. uvarum* CBS7001, *S. eubayanus* FM1318, *K. marxianus* DMKU31042, and *P. kudriavzevii* CBS573. Duplicated reads were marked with Picard 2.6.0[48], and local realignment around indels and variant calling were performed with GATK v4.1.1.0[49]. Allele balance information was incorporated in the VCF files with the GATK Variant Annotator. The called genotypes were filtered to keep only biallelic SNPs with a minimum depth of 5, a QUAL score of at least 30 and positions called in at least 90% of the isolates. Additionally, SNPs were also obtained by mapping the filtered reads only to the *S. cerevisiae* reference

genome to call variants taking into account introgressed segments (Supplementary Figs. 3 and 4).

## Multidimension scaling analysis

Genome-wide SNP data were used for multidimension scaling (MDS) analysis with PLINK v1.9[50]. A total of 1262 isolates were selected to provide a global overview at the species level, including 1011 from Peter et al.[6], 216 newly sequenced in this study, 33 strains from previous sequencing efforts in the Neotropics (Barbosa et al.[16], Barbosa et al.[40]) and four additional sequences from South America-associated lineages (SAM, Tellini et al.[12]). SRA Toolkit v3.0.0 was used to retrieve FASTQ files corresponding to samples other than those sequenced in this manuscript. Two sequences were excluded from Fig. 1a due to redundancy with other entries detected with PLINK[50]. Only nuclear SNPs were considered, and low-frequency or rare variants were excluded from this analysis using a threshold of missing call rate > 1%. After the application of this filter, isolates with missing genotypes > 10% were discarded. The quantitative indices (components) of the genetic variation for each isolate were calculated on the genome-wide average proportion of alleles shared between any pair of individuals within the sample. To perform the MDS analysis with equal representation across clades (Supplementary Fig. 2), we followed the same steps described above, but using only up to five randomly selected isolates from each phylogenetic clade described in the following section.

## Phylogenetic analysis

Maximum-likelihood trees were constructed using genomes sequenced in this study along with global reference strains, with particular emphasis on the American clades. Specifically, in the supplementary complete phylogenetic analyses (Supplementary Fig. 1), we included 487 strains distributed as follows: all 216 newly sequenced strains generated in this study, 23 sequences from Barbosa et al.[16], 10 isolates from Barbosa et al.[40], four South American isolates (SAM) from Tellini et al.[12] and at least one representative of each clade defined by Peter et al.[6] ($n = 213$). To improve resolution within lineages containing agave-associated isolates, we deliberately overrepresented strains from the Mixed Origin, Wine, and North American Oak clades from Peter et al.[6]. Additionally, we incorporated 21 Alpechin isolates from Pontes et al.[37], as a reference to determine the effects of extensive introgression content. The 487 strains are indicated in the column *Used_in_Supplementary_Complete_Phylogenies* of Supplementary Data 2. For the primary phylogeny depicted in Fig. 1b, we used the same strain set but excluded admixed individuals. Only 332 strains exhibiting > 85% of a predominant ancestral component as inferred from the ADMIXTURE analysis were retained for this tree. The 332 strains are indicated in the column *Used_in_Main_Phylogeny* of Supplementary Data 2.

To generate the phylogenies, haplotype sequences were obtained with vcf2phylip v2.3[51] from the nuclear genomic variants that mapped to the *S. cerevisiae* subgenome from the alignment to the concatenated species genome or, when indicated, directly to the *S. cerevisiae* reference genome to include variants from introgressions. The phylogenetic trees were inferred using IQTree[52] 2.3.6 with the best model calculated with −m MFP and adding the + ASC parameter with 1000 ultrafast bootstraps and aLRT (aLRT shown for key branchings in Supplementary Fig. 2). Phylogenies were also reconstructed using RAxML v8.2.12[53] (raxmlHPC-PTHREADS-AVX2) with the GTR + GAMMA model and 100 bootstrap replicates. Since the results were similar with the two methods, only IQTree trees were used for the figures of the manuscript. The visualization of the trees was performed using Microreact v.282[54]. To generate the trees with equivalent number of strains per clade (Supplementary Fig. 2), we randomly selected up to five isolates from the phylogenetic clades included in the set of 332

genomes and built the phylogeny with IQTree 2.3.6 with the ASC parameter.

## ADMIXTURE

For the ADMIXTURE analysis, we analyzed a dataset comprising 466 strains centered on the newly sequenced agave-associated isolates, as indicated in the column *Used_in_ADMIXTURE* of Supplementary Data 2. This included all strains sequenced in the present study ($n = 216$), along with 213 representative genomes of strains from Peter et al.[6] ensuring coverage of all clades and inclusion of 20 strains without clade assignment. In addition, to capture phylogenetic and introgression diversity, we incorporated 23 strains from Barbosa et al.[16], 10 strains from Barbosa et al.[40] and 4 strains reported by Tellini et al.[12] that belong to the SAM lineages. To enhance resolution within clades where agave-associated strains clustered, we intentionally overrepresented the Mixed Origin, Wine, and North American Oak lineages in the selection from Peter et al.[6]. To run ADMIXTURE 1.3.0[55], the VCF with 466 genomes was curated using PLINK 1.9[50] to remove linkage disequilibrium with parameters previously used for *S. cerevisiae*[56] (50 window size, sliding every 5 nucleotides and correlation of 0.5). Curation led to a final set of 618,253 SNVs. Ten seeds were used to run ADMIXTURE. The cross-validation (CV) error was obtained for models with $K = 2$ to $K = 35$. The lowest median CV error $K$ was 24, and therefore it was used for Fig. 3a. Visualization was made in Pong 1.4.9[57]. The R package vegan (2.7-2)[58] was used to perform the Spearman Mantel test.

## Estimation of genetic diversity

To calculate genetic diversity (π) and Tajima's $D$, we used VCFtools 0.1.14[59] and the VCF with variants that mapped to *S. cerevisiae* after mapping to the concatenated reference and filtering high-quality biallelic SNPs with less than 10% of missing data. Per variant heterozygosity was estimated using BCFtools 1.9[60] on individual strain VCFs to calculate the ratio of the number of heterozygous variants by the number of variants.

## Analysis of introgressions

To identify introgressed genes, we obtained the regions that mapped to the *S. paradoxus* genome when the alignment was performed to the concatenated reference. A gene was considered introgressed if over half of it had a coverage depth exceeding 25% of the median depth of the entire reference. Additionally, it had to be the ortholog with the highest depth or, if not, its depth had to be at least 25% of that of its ortholog in the *S. cerevisiae* subgenome. A minimum 5% nucleotide-level difference between the orthologs of *S. cerevisiae* and *S. paradoxus* was also a requisite. An introgressed gene was considered heterozygous if the ratio of the depth coverage between the two subgenomes in the competitive mapping was between 0.25 and 4.

After passing these mapping thresholds at the gene level, we grouped consecutive genes in blocks according to the annotation[61] and assessed the phylogenetic position of each block with respect to a panel of two *S. cerevisiae* (Wine ERS1082532 and Taiwanese ERS1082750) and five *S. paradoxus* (*SpA* SRR4074385, *SpC* SRR7500262, *SpB* SRR4074411, *SpB_Bra* SRR4074412, *SpB_MxAgave* YMX005537) genomes with IQtree. Using ape 5.0[62], we only kept as true introgressions those blocks that grouped with *S. paradoxus* in the phylogeny.

Gene Ontology enrichment analysis was performed using the GO term finder tool from the Saccharomyces Genome Database. Introgressed genes present in at least 75% of strains within each clade were used as the query set, with no background set specified. Using the default parameters, no enriched terms were found or were too broad to provide any meaningful biological insights.

Once the introgressed blocks were determined, to infer the origin of introgressions, we mapped the full panel of *S. paradoxus* genomes

from López-Gallegos et al.[23] to the concatenated reference genome that includes both *S. cerevisiae* and *S. paradoxus*. We retained only the *S. paradoxus* subgenome. We filtered the resulting VCF to retain only regions previously identified as introgressed from *S. paradoxus* into *S. cerevisiae* strains. Using this VCF containing introgressed regions in *S. cerevisiae* and the corresponding *S. paradoxus* variants, we calculated pairwise identity-by-state (IBS) values using PLINK v1.9[50]. For each introgressed region, we assigned its likely origin by identifying the *S. paradoxus* clade containing the strain with the highest IBS. When introgressed regions showed equally high similarity (IBS) to strains from multiple *S. paradoxus* lineages, no origin was assigned. We report the origin of introgressed regions as determined by the closest clade based on IBS similarity, as this approach leverages the most extensive dataset, including all *S. paradoxus* strains from López-Gallegos et al.[23], and provides improved resolution for assigning the origin of a larger number of introgressions. This assignment is concordant with results obtained using alternative metrics, including identification of phylogenetic siblings and assignment to the clade of the strain with the shortest phylogenetic distance (Supplementary Data 4, columns *ClosestClade_by_IBS_Similarity*, *Siblings_in_Phylogeny*, and *Strain_with_LowestDistance_in_Phylogeny*). All the genes within a given block were considered to be from the same origin.

## Map generation
Maps were generated in R using the ggplot2 4.0.0[63]. World coastline and political boundary data were obtained from the Natural Earth dataset via the rnaturalearth package 1.1.0[64].

## Reporting summary
Further information on research design is available in the Nature Portfolio Reporting Summary linked to this article.

## Data availability
Genome sequencing data generated in this study have been deposited in the NCBI SRA under the BioProject accession PRJNA1138754 [https://www.ncbi.nlm.nih.gov/bioproject/?term=PRJNA1138754]. The accession numbers of each genome employed in this study, including those previously sequenced, are provided in Supplementary Data 1 and 2. A Spanish translation of the article is provided in Supplementary Data 5. Source data are provided with this paper.

## Code availability
Scripts for identifying introgressed genes, determining their origins and generating the figures presented in this study are available at: Zenodo https://doi.org/10.5281/zenodo.17970285 (2025)[65].

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

## Acknowledgements

We thank Luis Aguilar (LIIGH-LAVIS), Porfirio Gallegos (Cinvestav), Aaron de Luna (LIIGH), Alejandra Castillo (LIIGH), Carina Uribe (LIIGH), Maritrini Colón-González (LIIGH) and Jair García (LIIGH) for technical assistance. We thank Bernard Dujon and Maria Ávila for critical reading of the manuscript. We thank Gianni Liti, Luis Delaye, Diego Ortega, Marcela Sandoval, Nicolò Tellini, and Alicia Mastretta-Yanes for helpful discussions, and Manuel R Kirchmayr (CIATEJ), Anne Gschaedler (CIATEJ), Maritza Álvarez (CIAD) and Marc-André Lachance for sharing strains. We are deeply grateful to all the producers of agave spirits and fermented beverages in Mexico who kindly participated in this work by providing access to fermentation samples and for sharing their knowledge. Details about the acknowledgements regarding help in field work and sampling of the YeastGenomesMx Consortium can be found in Gallegos Casillas et al.[20]. This work was funded by: Secretaría de Ciencia, Humanidades, Tecnología e Innovación de México (Secihti) (grants: FORDECYT-PRO-NACES/103000/2020 [A.D.L., L.M.], CF-2023-G-695 [E.M., L.M.], CBF-2025-G-838 [E.M., A.D.L., L.M.]); Fondo de Investigación y Desarrollo Tecnológico del Cinvestav (SEP-CINVESTAV/023, [A.D.L.]); UNAM-PAPIIT (grants: IN209021[L.M.], IN230420[L.M.] and IN212524 [L.M.]); UNAM-UI System Seed Grants 2023 [L.M.]; and the UK BBSRC under the

Global Challenges Research Fund (GCRF) Growing Research Capability call through the CABANA Innovation Fund (BB/PO27849/1 [E.M., A.D.L.]). E.M. was funded by Secihti for a sabbatical stay (IO2OO/111/2024). L.F.G-O. was funded by Secihti at the postdoctoral level (4133922), and I.S., J.A.U.A., and C.L-G. at the graduate level.

## Author contributions

Conceptualization: E.M., A.D.L., and L.M.; methodology: J.A.A-R., I.S., L.F.G-O., J.A.U.A., C.L-G. and X.A-D.; data curation: J.A.A-R., I.S., and L.F.G-O.; investigation: C.L-G; formal analysis: J.A.A-R., I.S., L.F.G-O., J.A.U.A., C.L-G., and X.A-D.; Visualization: J.A.A-R., and I.S.; funding acquisition: E.M., A.D.L., and L.M.; supervision: E.M., A.D.L., and L.M.; writing original draft: J.A.A-R., E.M., A.D.L. and L.M.; writing, review and editing: all authors.

## Competing interests

The authors declare no competing interests.

## Additional information

[1]Unidad de Genómica Avanzada, Centro de Investigación y de Estudios Avanzados del Instituto Politécnico Nacional, Irapuato, Mexico. [2]Laboratorio Internacional de Investigación sobre el Genoma Humano (LIIGH), Universidad Nacional Autónoma de México, Querétaro, Mexico. [3]Posgrado en Ciencias Biológicas, Universidad Nacional Autónoma de México, CDMX, Mexico. [4]Unidad Irapuato, Departamento de Ingeniería Genética, Centro de Investigación y de Estudios Avanzados del Instituto Politécnico Nacional, Irapuato, Mexico. [5]Investigadoras e Investigadores por México, Secretaría de Ciencia, Humanidades, Tecnología e Innovación, CDMX, Mexico. [6]Centro de Investigación sobre el Envejecimiento, Centro de Investigación y de Estudios Avanzados del Instituto Politécnico Nacional, CDMX, Mexico. [7]Present address: IRCAN, INSERM, CNRS, Côte d'Azur University, Nice, France. [8]Present address: Department of Human Genetics, University of Chicago, Chicago, IL, USA. [9]These authors jointly supervised this work: Eugenio Mancera, Alexander DeLuna, Lucia Morales. ✉e-mail: eugenio.mancera@cinvestav.mx; alexander.deluna@cinvestav.mx; lmorales@liigh.unam.mx

