## [Transparent Peer Review File · Nature Communications]

Recurrent introgression and geographical stratification shape *Saccharomyces cerevisiae* in the Neotropics

Corresponding Author: Dr Lucia Morales

Version 0:

Reviewer comments:

Reviewer #1

(Remarks to the Author)

Avelar-Rivas et al present a study into the genomic characteristics of *Saccharomyces cerevisiae* strains isolated from the Neotropics via re-sequencing of 216 isolates. They define a new clade "Mexican Agave 2" and map numerous introgressions from *Saccharomyces paradoxus* across the genomes of many of neotropical strains.

The expanded repertoire of isolates has expanded the catalogue of strains from this region. However the focus on the widespread introgressions from *S. paradoxus* largely recapitulates findings from previous large-scale de novo sequencing efforts, such as those of Peter et al 2018 and D'Angiolo et al 2020, which defined the Alpecin, Mexican Agave and French Guiana lineages as displaying high degree of introgression from this sister species.

Given that the authors have used a mapping-based approach to defining introgression, more detailed explanation of the methodology (with figures where appropriate) would be useful for determining the robustness of the predictions relative to the a gene-prediction + phylogenomics approach. Direct comparison to data from papers such as Peter et al 2018 could be used to ground-truth the mapping-based methodology.

Likewise, the application of introgressed gene blocks to specific *S. paradoxus* strain origin should be supported by a gene-wise phylogenetic approach rather than the distance-based methodology applied.

Reviewer #2

(Remarks to the Author)

This relevant study is based on the investigation of circa 200 isolates of *S. cerevisiae* collected from agave fermentations in Mexico and is centered on the characterization of multiple and independent introgressions originating in the sister species *S. paradoxus*. I find this investigation well conducted and significant as the natural history and domestication of *S. cerevisiae* in the Neotropics is poorly understood. Moreover, the pervasiveness of *S. paradoxus* introgressions in *S. cerevisiae* found in the Central and South America, either in wild or in domesticated populations, is unique and understanding its causes is of relevance as wild populations in other regions lack such introgressions. Although this study does not solve the question of why are introgressions found in Neotropics but not in other regions, it is a significant contribution to the topic. Below I list my comments and questions that I hope the authors find pertinent.

Main

1. I wonder why the authors characterize the agave fermentations as "a habitat at the interface of natural and industrial environments" (lines 8-9). What are exactly the characteristics of this fermentation that separate it from other fermentations? For example, are spontaneous wine fermentations also at this interface? If so, is wine must a natural substrate? I believe that the distinction between wild and artificial environments should be made very clear from the onset of the paper, and I see no reason not to consider agave fermentations as a typical artificial environment created by humans. Also, why using "habitat" in this context (see also line 24)? In order to demarcate clearly wild and artificial substrates, perhaps "habitat" could be restricted to the context of natural settings and a more neutral expression like "environment" could be used in situations relating to artificial man-made environments.

2. The authors use the expression "neotropical cluster" (line 37) for designate the three *S. cerevisiae* lineages from this

region. I agree that treating these lineages like this makes sense, but I miss a more thorough dissection of this group. I believe it would be important to emphasize (or at least mention) that two of these lineages are domesticated and one (Wild Brazil 3) is natural. That is relevant for several reasons, most importantly for discussing the occurrence of introgressions both in wild and in domesticated lineages. To add to this, the authors even state (line 94) "...distinguishing strains from the Neotropics (blue) from wild (green) and domesticated (red)", implying that the strains from the Neotropics are neither wild or domesticated, which is odd.

3. The species "evolutionary history" needs to be studied on wild isolates, which is not the case in this study. I recommend reframing the way the study is presented as discussed. The following sentence (line 38) exemplifies my point "Considering the vast area encompassed by this region, the few available sequenced isolates likely provide an underestimate of genetic diversity, thus limiting our understanding of the species evolutionary history in this part of the world".

4. In lines 34-38 (see also line 86) the authors introduce three previously recognized neotropical lineages, that they refer to as Mexican Agave (MA), French Guiana (FG), and South American Mix 2 (SAM2). This last designation corresponds to an earlier designation, Wild Brazil 3 (see reference 17). I wonder why the authors preferred to use a designation that is not only superfluous but also confusing. In fact, "Mix" is normally used in the context of *S. cerevisiae* populations to label admixed lineages (see reference 6 for several examples). It is therefore unfortunate that the authors choose to associate a name that implies admixture to a non-admixed population. This confusion is even exposed by the authors in the construction and legend of Fig 1C in which the Wild Brazil 3 (SAM2) population is treated as admixed (see also lines 116, 122), which is false.

5. I understand the interest in linking *S. paradoxus* with agave fermentations, but when checking reference 18 it is difficult to find evidence for this. The reality is that for more than 1500 *S. cerevisiae* isolates one a single *S. paradoxus* isolate was found. I think it is difficult to argue that *S. paradoxus* is present in agave fermentations and I think this part need to be turned down. Moreover, is reference 19 adequate in this context?

6. Fig.1A reports on 1264 genomes but table S1 lists only the 261 agave fermentation genomes. Moreover, Fig 1C reports on 464 genomes. If they are included in table S2 this should be made clear by identifying which genomes were used in each analysis. Moreover, how were these different subsets chosen? Was there a preoccupation in representing all known populations? Why so many wine strains? Care much be applied because population structure analyses can be distorted by over-representing / neglecting individuals from different populations. The separation seen in figure 1A appears exacerbated and in fact cannot be confirmed. I would suggest that it oversimplifies the problem by showing a continuum from wine to wild (China?) isolates, with agave isolates as the only ones escaping this continuum. Moreover, the text in line 88 "The rich diversity of the newly sequenced strains is evident, as strains from the tropical Americas encompass nearly half of the global diversity" appears to ignore the fact that most diversity resides in wild oak-associated strains and that the agave strains (either those already known or the new ones, are domesticated and therefore are not expected to be considerably diverse). I am convinced that a less spectacular but probably more truly representation would not diminish the current study (and this is true for the entire perspective of the paper).

7. Fig. 1B. This representation might the consequence of the sampling with a limited number of wild strains (again, no information on the origin of the strains used), a single known domesticated group (wine but not beer1 and 2, bread, sake, Asian beverages, etc) and an over-representation of agave strains.

8. Fig 1C. Again, I wonder if the phylogeny is representative of the known phylogeny of the species, i.e. includes the relevant lineages. My point is that only the MA2 is a new addition, and previous phylogenies did not show the MA1 and FG in such a distant position from the rest of the species lineages.

9. Line 293. This new population or strain SpB_Mx needs to be presented. The reader also needs to be introduced to the A, B and B* populations.

!0. The Wild Brazil 3 population is missing in the study of Fig. 4.

Minor

1. In this sentence (line 36-37) "Notoriously, the handful of sequenced genomes available to date indicate that this Neotropical cluster harbors an unusually high and heterogeneous number of introgressions from *S. paradoxus*", the first study revealing these introgressions (ref 17) is not mentioned.

2. Populations should be designated in a consistent manner throughout the paper – line 91 wild North American"; line (119) "wild North America Oak clade";

3. In Figure 1C and in the text what is the relevance in separating the Mexican agave strains in two subgroups? Wouldn't it be preferable presenting the strains from this fermentation as a single group, given its technological utilization, etc? For future studies having a single population recognized as originating from Agave fermentations might be advantageous. Think of the "wine" population and of what it represents for the understanding of *S. cerevisiae* biology and of the irrelevance of subdividing it (currently being attempted by several authors).

4. In line 148 "...revealing that what was previously described as "Mexican Agave" represents only a 148 minor fraction of

the great genetic diversity found in agave fermentations” this statement appears to be too strong (see comments above) as the new agave strains are numerous but not vastly different from those already known.

5. Figure 2. Why $K=24$?

6. Line 194 “Likewise, there were SAM2 strains showing genetic components of the MA2 and French Guiana clades”. This is not surprising since SAM2 (Wild Brazil 3) is the closest wild relative of the domesticated MA2 and French Guiana clades. This could be added to the sentence.

7. Line 204-205. Same for WB3.

8. Line 233-235. I could not follow the reasoning as it appears to contradict what was stated in the lines before. Please clarify why this cannot be a recent introgression event.

9. Line 279. The alpechin clade has a totally distinct introgression history that occurred in Europe and involved European *S. paradoxus*. Thus, it appears not relevant for this comparison.

10. Line 294. “not commonly found” or not found at all?

11. Line 320. With respect to anthropogenic environments the authors appear to have missed this study <https://doi.org/10.1093/gbe/evy132>.

12. Line 327. Is it possible to conceive an alternative scenario in which there was a single pulse but MA1 did not lose as many *paradoxus* genes as MA2?

13. Line 331. Was any attempt made to investigate the pool of introgressed genes with respect to function?

14. Line 342. I think it is important to focus on sampling natural environments rather than anthropic ones (see previous comments on how to address the “true” natural history, rather than the recent domestication history).

15. Line 349. Was the actual genetic divergence calculated?

16. Line 397-398. Why not use IQ-TREE with the ‘ASC’ parameter associated with the best model inference? This takes into account that the dataset is SNPs and not coding sequences/proteins.

Reviewer #3

(Remarks to the Author)

Version 1:

Reviewer comments:

Reviewer #1

(Remarks to the Author)

The authors have addressed my previous comments and have improved the manuscript significantly.

With the re-review of the material, I have a question regarding the models proposed by the authors regarding the formation of the MA1 genotype(s).

As pointed out by the authors, the strains within the MA1 clade show far higher variation in the specific introgressed genes than strains within other clades.

How do the authors propose that this variation in gene content in the MA1 cluster occurred? Would a full hybridisation *cerevisiae* x *paradoxus* event have occurred followed by differential gene loss? Are the various strains still undergoing genome reduction? Could this be a continual process with hybridisation commonly occurring?

Reviewer #2

(Remarks to the Author)

I was very pleased to see that the authors carried out a thorough and very careful revision of the manuscript.

As far the concerns I raised in my first revision, I see that the authors addressed them, and I do not have any relevant request for change in the manuscript.

There is however one issue, that of the divergence of the Neotropical cluster (Fig. 1B), in which I tend to disagree with the authors. I do not think that further scrutiny will support this divergent placing of the Neotropical cluster that is probably due to a unbalanced sampling of the Neotropical representatives, and perhaps of other groups like the wine clade, by comparison with the representatives of other populations. It is not the number, e.g. phylogenies with more than 1000 or 3000 genomes, but the fine balance of the numbers of genomes of representatives of ALL the populations that is of importance. Please check here <https://doi.org/10.1016/j.cub.2025.04.056> how a probably more balanced sampling (that of course misses the "Mexican Agave 2") fails to show the neotropical cluster as the utterly divergent lineage seen in Fig. 1. However, I want to stress that I am making this comment "for the record" not as a request for change.

I conclude by congratulating the authors for this very interesting and finely prepared manuscript that I am sure will stand as a relevant contribution for *S. cerevisiae* population biology.

Minor notes

Line 8, 102, 172, 224, 341, 401, 575 "open agave fermentation". Why not employing "spontaneous fermentation" which provides the reader with an objective concept, clearer than that of "open fermentation" ?

Line 30 "The Neotropical region, which extends from..."

Line 97 Agave fermentation strains belong TO a genetically diverse Neotropical cluster

Reviewer #3

(Remarks to the Author)

July 2025

ITEMIZED RESPONSE TO REVIEWERS

Manuscript ID NCOMMS-24-72197-T

Title: "Recurrent introgression and geographical stratification shape *Saccharomyces cerevisiae* in the Neotropics" by Avelar-Rivas et al.

We thank the editor and the three reviewers for their thorough evaluation and constructive feedback. We were encouraged to learn that our findings were found relevant and that a revised version of our manuscript may be suitable for publication. While the main findings of the study remain unchanged, specifically the identification of recurrent introgression events from *S. paradoxus* in distinct *S. cerevisiae* lineages, these conclusions are now more strongly supported through expanded analyses, clearer methodological descriptions, more cautious interpretation of results, and improved contextualization. These analyses required the contributions of two new members of our groups that are now included as co-authors. Overall, the integration of the reviewers' suggestions has resulted in a more rigorous and clearly presented manuscript.

The revised manuscript includes the following major changes:

1. Multiple down sampling analyses to address potential sampling biases in MDS and phylogenetic reconstructions, consistently confirming the phylogenetic position of the Neotropical cluster. These results are now included in the revised manuscript and new panels in **Figure S2**.

2. Incorporation of new analysis showing SNV counts relative to the reference genome across strains, to more accurately report the divergence and genetic diversity observed in the Neotropical cluster, which highlights that Neotropical strains rank among the most divergent after wild isolates from Taiwan and China (new **Figure 2** and new **Figure S3**).

3. Better description of the mapping-based pipeline and incorporated a phylogenetic validation step. To identify the origin of introgressed segments, we used identity-by-sequence (IBS) to determine the most similar potential donor, leveraging the full panel of *S. paradoxus* strains to maximize resolution (**Figure 5A**, **Data S1**).

4. A more detailed characterization of the set of introgressed genes among the different clades, emphasizing the patterns that support an independent introgression pulse in MA1 (**Table S3** and **Figure S9**).

5. A more thorough description of the composition of strain sets used in each analysis (**Figure 1**, **Table S1**, **Table S2**).

6. Reviewer #2 raised relevant concerns about the reference genomes used to trace the origins of *S. paradoxus* introgressions. We note that, during the time of this revision, the results and

genome data from a separate project by our group (focused on the ecology of *Saccharomyces* strains in wild and fermentative environments, including 58 sequenced strains of *S. paradoxus* from Mexico) are now publicly available as a preprint on bioRxiv. Reference to this work provides a stronger and clearer context for our revised manuscript.

See doi: <https://doi.org/10.1101/2025.05.31.656962>.

7. All remarks were carefully addressed and, in most cases, incorporated into the revised manuscript. In the few instances where changes were not made, we provide a clear rationale for why the original version was kept.

Below is a point-by-point response to the reviewers' comments. The page and line numbers in the responses refer to the version of the manuscript with tracked changes (AgaveYeast_ReviewTrackedChanges.pdf) to facilitate identification of the revisions. Minor corrections made during the revision, but not specifically requested by reviewers, were not marked using track changes.

Editor

Dear Dr Morales,

Thank you again for submitting your manuscript "Recurrent introgression and geographical stratification shape *Saccharomyces cerevisiae* in the Neotropics" to Nature Communications. We have now received reports from 2 reviewers (one with an Early Career co-reviewer) and, after careful consideration, we have decided to invite a major revision of the manuscript.

As you will see from the reports copied below, the reviewers raise important concerns. We find that these concerns limit the strength of the study, and therefore we ask you to address them with additional work. Without substantial revisions, we will be unlikely to send the paper back to review.

If you feel that you are able to comprehensively address the reviewers' concerns, please provide a point-by-point response to these comments along with your revision. Please show all changes in the manuscript text file with track changes or colour highlighting. If you are unable to address specific reviewer requests or find any points invalid, please explain why in the point-by-point response.

Important: In addition to the above, you must comply with the following editorial requests; we will not be able to proceed with your revised manuscript otherwise. Please also see the Nature Communications formatting instructions, which you may find useful while preparing your revised manuscript.

Reviewer #1 (Remarks to the Author):

Avelar-Rivas et al present a study into the genomic characteristics of *Saccharomyces cerevisiae* strains isolated from the Neotropics via re-sequencing of 216 isolates. They define a new clade "Mexican Agave 2" and map numerous introgressions from *Saccharomyces paradoxus* across the genomes of many of neotropical strains.

Comment 1.1 The expanded repertoire of isolates has expanded the catalogue of strains from this region. However the focus on the widespread introgressions from *S. paradoxus* largely recapitulates findings from previous large-scale de novo sequencing efforts, such as those of Peter et al 2018 and D'Angiolo et al 2020, which defined the Alpechin, Mexican Agave and French Guiana lineages as displaying high degree of introgression from this sister species.

We agree with the reviewer that prior studies have reported introgressions from *S. paradoxus* in certain *S. cerevisiae* lineages. However, we argue that our study is clearly distinct in showing multiple, independent introgression events within a geographically structured population. To our knowledge, this pattern has not been previously documented, likely because it is not common and it required high-resolution genomic data from numerous strains of both admixing species, sampled within the same geographic region. In contrast to the work of Peter et al. (2018) and D'Angiolo et al. (2020), which identified the presence and broad evolutionary origins of introgressed lineages in Europe (Alpechin) and the Americas, our study provides a novel contribution by analyzing in detail the genomic population structure of such introgressed lineages. Specifically, with these analyses we show that, in the Neotropical clade, introgressions are not the result of a single hybridization event—such as that proposed for the origin of the highly introgressed Alpechin lineage—but instead originate from multiple, recurrent hybridization events with *S. paradoxus*.

To clarify this point, we have revised the Introduction (**lines 45–58** in the tracked-changes file provided) better explaining previous work and highlighting the novel contributions of our study regarding introgressions and geographical structure. This revised section includes references to the studies of Peter et al. (2018) and D'Angiolo et al. (2020), as well as other studies that have investigated introgressions in *S. cerevisiae* (Barbosa 2016, Barbosa 2018, Tellini 2024).

Comment 1.2 Given that the authors have used a mapping-based approach to defining introgression, more detailed explanation of the methodology (with figures where appropriate) would be useful for determining the robustness of the predictions relative to the a gene-prediction + phylogenomics approach. Direct comparison to data from papers such as Peter et al 2018 could be used to ground-truth the mapping-based methodology.

As requested by the reviewer, our mapping-based pipeline is now better described in the "Analysis of Introgressions" section of the Methods (**lines 692–698**). Mapping-based strategies have been previously used for identifying introgressed fragments in (Barbosa et al 2016). The parameters specific to our workflow are reported in the Methods section.

In addition, we now define introgression by complementing the mapping-based approach with a phylogenetic analysis of the 13,711 identified regions. This phylogenetic analysis was applied to introgression blocks rather than to single genes, as the former provides greater size and variant density, and therefore higher resolution for the analysis. We think the integration of the phylogenetic analysis increases the robustness of our approach for identifying introgressions.

Also based on the reviewer's comment, we compared the introgressed genes identified in our analyses with those reported by the gene prediction and phylogenomics approach in Peter et al. (2018), as well as with genes identified by using *bona fide* diagnostic SNP markers that discriminate *S. cerevisiae* from *S. paradoxus*, as reported in Tellini et al. (2024). In our revised manuscript, we now show the overlap between our strategy for defining introgressions (based on the mapping approach combined with phylogenetic analysis of introgressed blocks), and the two signaled studies (lines 298–306, new Figure S7). Using 46 strains for which introgressions were defined by all three studies, we compiled a reference set of 3,662 genes ("Universe"), of which 1,582 were common to all three studies, representing 63.68%, 54.60%, and 74.13% of the total genes reported in this work, Tellini *et al.*, and Peter *et al.*, respectively. The new Figure S7 also shows a comparison in which we computed the overlap between our method and the strains reported in Barbosa *et al.* (2016), using the three strains shared between both datasets.

Supplementary Figure S7. Comparison of the method used in this study for detecting introgressed genes with alternative approaches. Venn diagrams show the number of genes identified as introgressed across 46 strains (left) and 3 strains (right) using three different identification approaches.

Comment 1.3 Likewise, the application of introgressed gene blocks to specific *S. paradoxus* strain origin should be supported by a gene-wise phylogenetic approach rather than the distance-based methodology applied.

To perform the phylogenetic approach suggested by the reviewer, we took advantage of the 13,711 phylogenies used to define introgressions (see response 1.2) and assigned the phylogenetic origin to each introgressed block when the sister sequence in the phylogeny was

also the one with the smallest phylogenetic distance. This analysis showed good correspondence with our previously reported results (see figures below, where (A) is our original result, and (B) is the result from the phylogenetic approach). In the phylogenetic approach, we identified the donor strain as the one with the lowest number of differences (i.e., smallest distance or greatest similarity) in a phylogeny of reconstructed sequences from a VCF file with 5 representative *S. paradoxus* strains, two *S. cerevisiae* strains as outgroups and the introgressed block. The use of 5 *S. paradoxus* strains in the phylogenetic approach facilitated the analysis of the topologies but lacked resolution, as many introgressed blocks could not be confidently assigned to a specific lineage. In such cases, we labeled the origin as “SpB_undefined” when the sister clade included two SpB strains or when the strain with the lowest phylogenetic distance was not the sister clade. Blocks that were sister to both SpB and SpC were assigned the label “Sp_American.”

For the main analysis in the revised article, we decided to keep a sequence similarity-based approach, taking advantage of the full panel of *S. paradoxus* donor strains. However, we only included introgressions that were independently validated by the phylogenetic approach (see response 1.2). In addition, to improve donor assignment, we calculated the identity for each introgressed block using PLINK, comparing them to the entire *S. paradoxus* panel. The results shown in the revised manuscript are consistent with those from both strategies described above, the original distance-based methodology and the phylogenetic approach proposed by the reviewer (new **Figure 5A**). In the main article we present only the improved similarity-based method, as it yields more precise assignments while supporting the same overall conclusions. We note that we also incorporated uncertainty in donor lineage assignment into the model, by highlighting introgressed blocks shared across all Neotropical cluster clades prior to full divergence of the SpB subgroups (modified **Figure 5B**).

A) Original results

B) New results using the phylogenetic approach (discussed but not included in the final manuscript)

Reviewer #2 (Remarks to the Author):

2 This relevant study is based on the investigation of circa 200 isolates of *S. cerevisiae* collected from agave fermentations in Mexico and is centered on the characterization of multiple and independent introgressions originating in the sister species *S. paradoxus*. I find this investigation well conducted and significant as the natural history and domestication of *S. cerevisiae* in the Neotropics is poorly understood. Moreover, the pervasiveness of *S. paradoxus* introgressions in *S. cerevisiae* found in Central and South America, either in wild or in domesticated populations, is unique and understanding its causes is of relevance as wild populations in other regions lack such introgressions. Although this study does not solve the question of why are introgressions found in Neotropics but not in other regions, it is a significant contribution to the topic. Below I list my comments and questions that I hope the authors find pertinent.

Main

Comment 2.1. I wonder why the authors characterize the agave fermentations as “a habitat at the interface of natural and industrial environments” (lines 8-9). What are exactly the characteristics of this fermentation that separate it from other fermentations? For example, are spontaneous fermentations also at this interface? If so, is wine must a natural substrate? I believe that the distinction between wild and artificial environments should be made very clear from the onset of the paper, and I see no reason not to consider agave fermentations as a typical artificial environment created by humans. Also, why using “habitat” in this context (see also line 24)? In order to demarcate clearly wild and artificial substrates, perhaps “habitat” could be restricted to the context of natural settings and a more neutral expression like “environment” could be used in situations relating to artificial man-made environments.

We agree with the reviewer that agave fermentations are similar to other “spontaneous” fermentations and have revised the manuscript to avoid terminology that could be misleading in

this regard (e.g., **line 9** in the tracked-changes file provided). Given that agave spirit producers do not intentionally inoculate these fermentations, the microorganisms involved likely originate from nearby natural environments. This is why we initially referred to them as occurring at the “interface of natural and industrial environments.” To clarify this point, we now explicitly state throughout the Introduction, Results, and Discussion that agave fermentations are an anthropogenic environment (**lines 83 and 490**). For instance: “*In this study, we sequenced 216 strains from open agave fermentations, providing an extensive population genomics dataset for isolates of anthropogenic environments in the Neotropics.*” (**line 490**). We also added a paragraph to the Discussion (**lines 556-569**) to emphasize the anthropogenic nature of the environment and to acknowledge the overlapping zones between anthropogenic and natural environments.

Likewise, we revised and modified the use of the term “habitat” in the few instances where it previously appeared (**lines 9, 25–26**).

Comment 2.2. The authors use the expression “neotropical cluster” (line 37) for designate the three *S. cerevisiae* lineages from this region. I agree that treating these lineages like this makes sense, but I miss a more thorough dissection of this group. I believe it would be important to emphasize (or at least mention) that two of these lineages are domesticated and one (Wild Brazil 3) is natural. That is relevant for several reasons, most importantly for discussing the occurrence of introgressions both in wild and in domesticated lineages. To add to this, the authors even state (line 94) “...distinguishing strains from the Neotropics (blue) from wild (green) and domesticated (red)”, implying that the strains from the Neotropics are neither wild nor domesticated, which is odd.

As suggested by the reviewer, we have included a more thorough dissection of the cluster by separating previously sequenced neotropical strains in four groups that now include the Wild Brazil 3 (WB3, **line 36**). We also now clarify which of these isolates come from anthropogenic environments and which are from natural areas (**line 38**). Importantly, the WB3 is now considered as a separate group in the new version of the phylogenetic analysis (see modified **Figure 1**) and, throughout the text, we acknowledge and discuss that introgressions occur in both wild and human-associated lineages.

This reviewer’s comment helped us recognize that our initial description of the Neotropical cluster—contrasting it with “wild” and “domesticated”—was inaccurate. These terms were poorly chosen and have now been removed, along with the ADMIXTURE panel originally shown in Figure 1. Our intention was to convey that, in the admixture analysis with $k = 3$, the Neotropical cluster represents one of the genetic components, while the other two correspond predominantly to wild strains (e.g., Wild Asian and North American Oak, as broadly defined in Loegler et al., 2024 and Han et al., 2021) and human-associated strains (e.g., Wine, Dairy). However, this framing was an oversimplification, as all three components include a mix of strains from wild and anthropogenic origins. The use of the terms “wild” and “domesticated” was therefore misleading and now has been thoroughly revised. Additionally, the overrepresentation of Neotropical strains

in the ADMIXTURE analysis may have biased the inferred separation at this scale (see response to comment 2.6). These issues have been addressed in the revised version of **Figure 1**.

Comment 2.3. The species “evolutionary history” needs to be studied on wild isolates, which is not the case in this study. I recommend reframing the way the study is presented as discussed. The following sentence (line 38) exemplifies my point “Considering the vast area encompassed by this region, the few available sequenced isolates likely provide an underestimate of genetic diversity, thus limiting our understanding of the species evolutionary history in this part of the world”.

We acknowledge that the species’ evolutionary history is a subject of ongoing efforts and that wild isolates are essential to ultimately address this question. While our study focuses on anthropogenic environments, we believe the results contribute to understanding the broader natural history of *S. cerevisiae* in the Neotropics, consistent with perspectives in previous studies (e.g. Duan et al., 2018 and De Chiara et al., 2022).

In response to the reviewer’s comment, we have made the following changes to the manuscript: First, we have revised wording in the Introduction section to avoid broad claims about “evolutionary history” (**line 43, line 63**). In addition, we now explicitly note the need for additional sampling from environmental sampling (Discussion, **line 484-487, 532**). In the Discussion (starting at **lines 572-574**), we clarify that the observed genetic diversity could reflect evolutionary processes in both natural and human-associated contexts. Finally, we added the following sentence to the Discussion: “*Comprehensive sequencing of S. cerevisiae genomes in megadiverse regions is essential to understand the genomic diversity of this species, its natural evolutionary history, and its relationship with domestication.*” (**lines 484-487**), explicitly separating both concepts. These revisions provide a more cautious and precise framing of our study’s contributions, in line with the reviewer’s comment.

Comment 2.4. In lines 34-38 (see also line 86) the authors introduce three previously recognized neotropical lineages, that they refer to as Mexican Agave (MA), French Guiana (FG), and South American Mix 2 (SAM2). This last designation corresponds to an earlier designation, Wild Brazil 3 (see reference 17). I wonder why the authors preferred to use a designation that is not only superfluous but also confusing. In fact, “Mix” is normally used in the context of *S. cerevisiae* populations to label admixed lineages (see reference 6 [Peter et al. 2018] for several examples). It is therefore unfortunate that the authors choose to associate a name that implies admixture to a non-admixed population. This confusion is even exposed by the authors in the construction and legend of Fig 1C in which the Wild Brazil 3 (SAM2) population is treated as admixed (see also lines 116, 122), which is false.

We thank the reviewer for this clarification. We acknowledge that our initial labeling of strains as “SAM2” and its interpretation as a previously recognized lineage oversimplified the original definitions. In particular, we did not fully account for the distinction between the Wild Brazil 3 population described by Barbosa et al. (2016) and the admixed strains assembled as SAM2 in

Tellini et al. (2024). We now address this distinction more carefully and in line with the reviewer's observations.

As described in the response to comment 2.2, we have separated the three Wild Brazil strains from Barbosa et al. (2016) into a distinct group labeled Wild Brazil 3 (WB3; **see line 36**). This implied re-running most of the analyses accordingly. After this adjustment, the strains remaining in the SAM2 group (as defined by Tellini et al., 2024) display multiple genetic components at most K values in the ADMIXTURE analyses, consistent with the reviewer's definition of a population labeled "Mix" (see **Figure S4** and **Table S2**, columns Main_Ancestral_pop, Main_Anc_Pop_Q, Secondary_Ancestral_Pop, Secondary_Anc_Pop_Q). In the revised manuscript, all the admixed strains were removed for phylogenetic analyses, including three from SAM2, while the non-admixed WB3 strains are shown (see modified **Figure 1B**). These changes are also now reflected in the text: **Lines 36, 101-102, 131-136, 144-145, 156-157, 776, 793, and 10121**.

Comment 2.5. I understand the interest in linking *S. paradoxus* with agave fermentations, but when checking reference 18 (Gallegos-Casillas, 2024) it is difficult to find evidence for this. The reality is that for more than 1500 *S. cerevisiae* isolates one a single *S. paradoxus* isolate was found. I think it is difficult to argue that *S. paradoxus* is present in agave fermentations and I think this part needs to be turned down. Moreover, is reference 19 (Peris et al, 2023) adequate in this context?

Based on this comment, we have toned down the statement about the presence of *S. paradoxus* in agave fermentations, acknowledging that it is found at low frequency in this environment (**lines 55-58**). However, we note that the presence of *S. paradoxus* in agave-spirit distilleries has been consistently reported, and we now cite recent studies to support this claim in a much stronger manner. Specifically, we now reference a recent preprint by our group, showing that 7.5% of over 270 *Saccharomyces* isolates from fermenting tanks in agave-spirit distilleries correspond to *S. paradoxus* as based on MALDI-TOF species identification (López-Gallegos et al., 2025). We also reference a recent review (Colon-Gonzalez, 2025) which compiles reports of yeast communities in the agave fermentation environment. These studies make it clearer that *S. paradoxus* has a low but consistent presence in agave fermentations (see **line 57**).

We note that the reported isolation frequency of *S. paradoxus* in Gallegos-Casillas et al. (2024) is actually higher than one out of 1,500 isolates, but we fully recognize that this information is difficult to extract from the Supplementary Material of that manuscript. Heading 3.3 of the Results section of Gallegos-Casillas et al. (2024) ("Validation of MALDI-TOF biotyping by ITS sequencing") states: "Noteworthy, a small fraction (7.6%) of the putative *S. cerevisiae* isolates was assigned to its sister species *Saccharomyces paradoxus* by ITS sequencing."

In sum, to address the reviewer's well-taken point, we have replaced these citations in the revised manuscript with more recent studies (**line 55-58**), which we believe provide strong evidence that *S. paradoxus* is consistently present in agave fermentations, albeit at a much lower prevalence than its sister species.

Comment 2.6. Fig.1A reports on 1264 genomes but table S1 lists only the 261 agave fermentation genomes. Moreover, Fig 1C reports on 464 genomes. If they are included in table S2 this should be made clear by identifying which genomes were used in each analysis. Moreover, how were these different subsets chosen? Was there a preoccupation in representing all known populations? Why so many wine strains? Care much be applied because population structure analyses can be distorted by over-representing / neglecting individuals from different populations. The separation seen in figure 1A appears exacerbated and in fact cannot be confirmed. I would suggest that it oversimplifies the problem by showing a continuum from wine to wild (China?) isolates, with agave isolates as the only ones escaping this continuum. Moreover, the text in line 88 “The rich diversity of the newly sequenced strains is evident, as strains from the tropical Americas encompass nearly half of the global diversity” appears to ignore the fact that most diversity resides in wild oak-associated strains and that the agave strains (either those already known or the new ones, are domesticated and therefore are not expected to be considerably diverse). I am convinced that a less spectacular but probably more truly representation would not diminish the current study (and this is true for the entire perspective of the paper).

To provide a more structured and detailed response, we have organized our responses addressing the concerns raised in this comment under the following points:

Comment 2.6.1. Fig.1A reports on 1264 genomes but table S1 lists only the 261 [216] agave fermentation genomes. Moreover, Fig 1C reports on 464 genomes. If they are included in table S2 this should be made clear by identifying which genomes were used in each analysis.

We have reformatted the two Supplementary Tables to present the information more clearly regarding which strains were used in each analysis. **Table S1** now includes all 216 strains sequenced in this study, along with their associated metadata, while **Table S2** lists a total of 1,284 strains:

- The 216 strains sequenced in this study (**Table S1**)
- 1,010 strains from Peter et al. (2018)
- 37 strains from the Americas used in Tellini et al. 2024 (strains from Barbosa et al. 2016; Barbosa et al. 2018; Gallone et al. 2016; Legras et al. 2018)
- 21 strains of the Alpechin clade (Pontes et al. 2019).

Binary indicator columns in **Table S2** indicate which set of strains was used in each analysis: Used_in_MDSA, Used_in_main_phylogeny, Used_in_ADMIXTURE, Used_in_complete_phylogenies, and Used_in_downsamplMDSPhylogeny.

These designations are now presented in a “readme” sheet in **Table S2** and are also described in the Materials and Methods section (**lines 606-612, 623-637 and 658-667**).

Comment 2.6.2. Moreover, how were these different subsets chosen?

We selected strain sets that were most relevant to each analysis, balancing informative value with computational feasibility. We have included explanation of the justification of the strain set in the corresponding section of each analysis as explained below:

Line 605: For the Multidimensional Scaling Analysis (Used_in_MDSA), we used 1,261 sequences to have a global overview of the species. These included the 1,010 strains from (Peter et al. 2018), the Neotropical strains sequenced in this study (n=216) and the 37 strains from SAM clades (Tellini et al 2024) that include strains from Barbosa et al. (2016) (n=23) from Barbosa 2018 (n=10) Legras et al 2018 (n=1) and Gallone et al 2016 (n=3). Two sequences were excluded from the final image due to redundancy with other entries detected with PLINK. As a result, the MDSA plot provides a comprehensive global overview of the species' diversity.

Line 623-637: For the phylogenetic analysis we focused on the newly sequenced agave-associated strains and their closest relatives. In the supporting, complete phylogenetic analyses (**Figure S1**), we included all newly sequenced strains from this study, those from any SAM group from Tellini et al. (2024), at least one representative from each clade described in Peter et al. (2018) and 20 strains without clade designation (Used_in_complete_phylogenies). To improve resolution in the agave-associated strains, we intentionally overrepresented the Mixed Origin, Wine, and North American Oak clades (Peter et al. 2018) since some of our isolates clustered within these groups. We also incorporated the 21 Alpechin strains from Pontes et al. (2019) given its high number of introgressions. For the main phylogeny shown in **Figure 1**, we used the same strain set defined for the supporting trees but excluded admixed strains (Used_in_main_phylogeny). For this figure, only strains with a predominant ancestral component, as determined by the ADMIXTURE analysis, were retained (n=332). For the new **Figure S2E**, we downsampled randomly the set of 332 strains selecting up to five sequences per clade ten times, the iteration that is shown as a representative result is in the column Used_in_downsampleMDSPhylogeny (new **Figure S2A**).

Line 658-667: For the ADMIXTURE analysis (Used_in_ADMIXTURE), we employed 466 strains and also focused on the newly sequenced agave-associated strains. Accordingly, we included all strains generated in this study (n=216), representative strains from each lineage described by Peter et al. (2018) (n=213) including at least one strain from each clade and 20 strains without clade designation. We also added strains that were of interest due to their previous classification as American strains that could inform the genetic components of our newly sequenced strains including strains from Tellini et al (2024, SAM clades), Barbosa et al. (2016), Barbosa et al (2018), Gallone et al (2016) and Legras et al (2018). As in the phylogenetic analysis, to improve resolution of the groups where agave strains clustered, in the selection of the 213 strains from Peter et al. (2018) we overrepresented the Mixed Origin, Wine, and North American Oak clades, as a few of our newly sequenced isolates clustered within these groups.

Comment 2.6.3. Was there a preoccupation in representing all known populations? Care much be applied because population structure analyses can be distorted by over-representing / neglecting individuals from different populations.

We agree with the reviewer that population structure analyses can be biased by the overrepresentation or omission of specific populations. In response to this concern, we have removed the $k = 3$ ADMIXTURE from Figure 1, along with the corresponding conclusions (**line 110-115**), as it overstated the prominence of the Neotropical cluster and the groups containing our newly sequenced strains. The ADMIXTURE analysis now show in **Figure 3A** is focused specifically on the Neotropical cluster. As detailed above, it was performed using all newly sequenced isolates, other strains from the Americas and representative strains from other clades of the world, as a reference. The conclusions drawn from this new analysis only concern the population structure of the Neotropical cluster (section starting on **line 219**). These changes have led to a more balanced and accurate interpretation of the population structure, particularly regarding the agave-associated and other strains from the Neotropics.

Comment 2.6.4. Why so many wine strains?

This comment likely arose from our ambiguous use of the “Wine” label in the original version of Figure 1 and from insufficient clarity about which strains were included in each analysis. The label was intended merely as a reference point to indicate the location of selected wine strains, which represent the most abundant group in Peter et al. (2018). However, not all strains under that label were actually wine strains. In the revised manuscript, we have improved Figure 1 by including the labels of key clades mentioned in the text and by specifying in the figure legend that all relevant lineages of the species were included as shown in Table S2, thereby eliminating the source of the confusion. In addition, the exact composition of the strains used in each analysis is now clearly provided in the relevant sections of Material and Methods and in **Table S2**.

Comment 2.6.5: The separation seen in figure 1A appears exacerbated and in fact cannot be confirmed. I would suggest that it oversimplifies the problem by showing a continuum from wine to wild (China?) isolates, with agave isolates as the only ones escaping this continuum.

To address the reviewer’s comment and test whether the separation shown in **Figure 1A** could result from the overrepresentation of the neotropical strains, we repeated the MDS analysis ten times using a downsampled dataset limited to five strains per clade. As shown in a representative iteration, the neotropical cluster still separated from the rest of the strains, although the distance was more modest (new **Figure S2A**). It is also worthwhile noting that in the dataset that we used for the MDS, the Neotropical cluster ($n=250$) was represented similarly as the Wine group ($n=268$), but only the former dispersed throughout the MDS plot.

We now restrain from drawing strong claims about the magnitude of the diversity based on the MDS analysis (line **104–108**) and included our downsampling analysis in new **Figure S2**.

Comment 2.6.6: Moreover, the text in line 88 “The rich diversity of the newly sequenced strains is evident, as strains from the tropical Americas encompass nearly half of the global diversity” appears to ignore the fact that most diversity resides in wild oak-associated strains and that the agave strains (either those already known or the new ones, are domesticated and therefore are not expected to be considerably diverse). I am convinced that a less spectacular but probably more truly representation would not diminish the current study (and this is true for the entire perspective of the paper).

We have revised this section of the Results to avoid drawing conclusions about the magnitude of diversity based on the MDS analysis (**line 104–106**). While our data indicate that the Neotropical cluster represents a diverse *S. cerevisiae* lineage, it clearly does not account for half of the global diversity, as we had inadvertently suggested.

To further address this point, we now provide additional evidence highlighting the unusually high diversity of the Neotropical strains. In a new analysis, we report the number of SNVs per group relative to the reference genome, based on 487 strains (new **Figure 2**). As the reviewer correctly noted, wild strains from Taiwan and China clearly showed the highest median number of SNVs. However, notably, the next most divergent groups are those from the Neotropical cluster.

In addition, we used a reference free approach that showed that Neotropical cluster clades show less identity by sequence than most groups to clades such as CHNII, Asian Fermentation, etc. Again, excluding Taiwanese and CHNI strains (**Figure S3**). We also note that Loegler et al. (2024) also remarked the high divergence of Mexican Agave isolates (**line 212-215**).

In sum, the revised version offers a more descriptive picture of the relative divergence and diversity observed in the Neotropical lineage, avoiding overstatements or exaggerated conclusions.

Comment 2.7. Fig. 1B. This representation might the consequence of the sampling with a limited number of wild strains (again, no information on the origin of the strains used), a single known domesticated group (wine but not beer¹ and 2, bread, sake, Asian beverages, etc) and an over-representation of agave strains.

In response to this and related comments by the reviewer, we have removed the ADMIXTURE analysis originally shown in **Figure 1B**. As noted in our response to point 2.6.4, the “Wine” label was meant to indicate selected wine strains as a reference, though neighboring strains included Beer, Alpechin, among others. Likewise, strains from North American Oak, Asian Islands, Sake, and Asian beverages were present but appeared near the “Wild” label, which likely caused confusion. We have clarified the composition of the strains sets in the corresponding section of the Materials and Methods and in **Table S2**.

Comment 2.8. Fig 1C. Again, I wonder if the phylogeny is representative of the known phylogeny of the species, i.e. includes the relevant lineages. My point is that only the MA2 is a new addition,

and previous phylogenies did not show the MA1 and FG in such a distant position from the rest of the species lineages.

The previous phylogeny in Peter et al. (2018) did show that the lineage is well separated in relatively long branches, even though it included a much smaller proportion of strains from the neotropics. The same was observed in the more recent study by Loeger et al. (2024), which included a larger number of neotropical strains, specifically from agave fermentations. These reported phylogenies are shown below. To make this point clear, we briefly describe it in the revised manuscript (**line 176**).

To further explore the impact of strain selection in the phylogeny, we repeated the analysis ten times using random downsampling to a maximum of five strains per clade. An example including admixed strains is shown below, and a version excluding admixed strains is now provided in the new **Figure S2F-G**. The resulting topologies were consistent with that shown in **Figure 1** whether or not introgressed regions were included: the Neotropical clade consistently occupied a distant position. This analysis is now described in the revised manuscript (**line 172**) and the resulting phylogenies are included as **Figure S2**.

Comment 2.9. Line 293. This new population or strain SpB_Mx needs to be presented. The reader also needs to be introduced to the A, B and B* populations.

To introduce the SpB_MxAgave population we now provide reference to our recent preprint about the ecology of *S. paradoxus* isolates from agave fermentations and natural environments (López-Gallegos et al., 2025). This study provides a better context to describe the origins of introgressions from this sister species (**lines 55–58**). Likewise, we now present the *S. paradoxus* lineages used to identify the origins of introgressions in the main text (**lines 450–457**).

Comment 2.10. The Wild Brazil 3 population is missing in the study of Fig. 4.

As described above, we now include the Wild Brazil 3 (WB3) as a standalone group throughout the manuscript, which is also shown now in **Figure 5** (formerly Figure 4).

Minor

Comment 2.11. In this sentence (line 36-37) “Notoriously, the handful of sequenced genomes available to date indicate that this Neotropical cluster harbors an unusually high and heterogeneous number of introgressions from *S. paradoxus*”, the first study revealing these introgressions (ref 17 [Barbosa et al. 2016]) is not mentioned.

The reference has been added (**line 45**) and we have expanded the description of the findings in this study as they are directly relevant for our work (sentences starting in **line 50**).

Comment 2.12. Populations should be designated in a consistent manner throughout the paper – line 91 wild North American”; line (119) “wild North America Oak clade”;

We replaced the designation “Wild North American” to “North American Oak” (lines 148 and 260) and made sure we used population names consistently throughout the manuscript.

Comment 2.13. In Figure 1C and in the text what is the relevance in separating the Mexican agave strains in two subgroups? Wouldn't it be preferable presenting the strains from this fermentation as a single group, given its technological utilization, etc? For future studies having a single population recognized as originating from Agave fermentations might be advantageous. Think of the “wine” population and of what it represents for the understanding of *S. cerevisiae* biology and of the irrelevance of subdividing it (currently being attempted by several authors).

We appreciate the reviewer's perspective. After careful consideration, we decided to retain the MA1 and MA2 designations, which is justified and appropriate given the scope and findings of this study. The decision is based on the following arguments:

- i) Although MA1 and MA2 are phylogenetically related, they form two clearly distinct clades separated by a mountain range known to act as a natural biogeographic barrier. This geographic structuring is a key finding of our study.
- ii) MA1 showed a recent pulse of introgression from *S. paradoxus* that is absent in MA2, highlighting differences in hybridization history, which is another major result of our study.
- iii) The two groups differ in patterns of admixture within species; for example, only MA1 has strong signatures of admixture with the wild clade North American Oak.
- iv) MA1 is only found in one of the seven agave-spirit producing regions, while MA2 is distributed across six regions and represents the dominant group in traditional agave-spirit production. This distinction is relevant for understanding and preserving regional diversity.
- v) We now show that MA1 and MA2 have an $F_{st}=0.43$, which is higher than all of the F_{st} values involving SAM2 and Tequila vs MA1, MA2, and WB3 (see response to comment 2.20). Furthermore, the IBS between MA1 and MA2 is 0.97, comparable to that between other well-defined clades such as between Mediterranean Oak and Alpechin (**Figure S3**).

We agree with the reviewer that unnecessary subdivision of strain groups may be counterproductive and anticipate that the community will likely consider Mexican Agave strains as a single group in future global-scale studies of *S. cerevisiae* populations. In the Discussion section of the current manuscript, we do refer to them as “Mexican Agave clades” (line 524). However, for more detailed population-level analysis presented here, we find it useful to distinguish between the two Mexican Agave groups.

Comment 2.14. In line 148 “...revealing that what was previously described as “Mexican Agave” represents only a 148 minor fraction of the great genetic diversity found in agave fermentations”

this statement appears to be too strong (see comments above) as the new agave strains are numerous but not vastly different from those already known.

Agreed, we toned down this statement, as suggested (**line 207**).

Comment 2.15 Figure 2. Why K=24?

We used K=24 because it was the K value with the lowest cross-validation error (**Figure S4**). This has been clarified in the main text (**line 224**) and explicitly stated in the figure legend.

Comment 2.16 Line 194 “Likewise, there were SAM2 strains showing genetic components of the MA2 and French Guiana clades”. This is not surprising since SAM2 (Wild Brazil 3) is the closest wild relative of the domesticated MA2 and French Guiana clades. This could be added to the sentence.

As mentioned in our previous responses, the WB3 strains are now distinguished from SAM2 strains. We have adapted the text accordingly (**lines 227-229**).

Comment 2.17. Line 204-205. Same for WB3. Lines 204-205: “For instance, the MA1, MA2, and Tequila Distillery clades all show higher genetic diversity than yeasts from the Wine clade.”

In the revised version, population metrics are reported only for groups with more than ten isolates, as these metrics are highly sensitive to outlier and sampling biases (legend of **Figure 3**). This approach has also been adopted in previous studies analyzing large numbers of lineages and strains (e.g. Peter et al., 2018).

Comment 2.18. Line 233-235: “The presence of both alleles in several introgressions of the Tequila Distillery clade suggests admixture with a non-introgressed population, which may also explain the lower number of introgressed genes”. I could not follow the reasoning as it appears to contradict what was stated in the lines before. Please clarify why this cannot be a recent introgression event.

We have substantially revised and expanded the explanation of the observed introgression patterns in these strains in the section beginning on **line 294**.

We removed focus from the patterns in the Tequila Distillery strains and instead we focused on those that in our opinion provide stronger evidence for multiple introgression pulses, without the possible confounding effects of high admixture.

Comment 2.19. Line 279. [In terms of “private” introgressions—those present in a clade but absent in all others—we scored 18.9% in FG, 32.2% in MA2, and 58.6% in MA1 (Figure 3E), while most introgressed genes in the Alpechin reference clade (76.3%) were exclusively found in this

clade (Figure 3E)]. The alpechin clade has a totally distinct introgression history that occurred in Europe and involved European *S. paradoxus*. Thus, it appears not relevant for this comparison.

As suggested by the reviewer, we have removed the comparison with the Alpechin lineage when it was not useful (e.g. **Figure 5A**). We also deleted previous Figure 3C-D. We retained the Alpechin group as a reference for comparison when useful.

Comment 2.20. Line 294: “In contrast, the majority of the introgressed genes in the FG strains tracked to *SpB*, a different lineage of *S. paradoxus*, which is not commonly found in agave distillery contexts”. “not commonly found” or not found at all?

The *SpB* lineage has actually been recovered only from natural environments, not from agave fermentation samples. We have modified the sentence and included the reference to the recent preprint by López-Gallegos, et al. (2025) (**lines 465-468**).

Comment 2.21. Line 320. “In the Americas, several *S. cerevisiae* lineages feature high number of introgressions from *S. paradoxus*, both in wild and anthropogenic environments”. With respect to anthropogenic environments the authors appear to have missed this study <https://doi.org/10.1093/gbe/evy132>. <https://academic.oup.com/gbe/article/10/8/1939/5047776>

We have included the reference in the relevant sentence (**line 497**), as well as in other appropriate sections of the manuscript.

Comment 2.22. Line 327. “At least one additional, more recent pulse of introgression from the *SpB_Mx* lineage took place in the MA1 clade, which is supported by high numbers of large and heterozygous introgressions in some strains of this clade”. Is it possible to conceive an alternative scenario in which there was a single pulse but MA1 did not lose as many *paradoxus* genes as MA2?

This comment led to the inclusion of new **Figure S9**, which shows the patterns of persistently introgressed genes in the Neotropical clades, with particular emphasis on the large heterozygous regions present only in a small subset of MA1 strains. Considering the phylogenetic history of these strains, explaining the observed patterns without invoking a second introgression event in this clade would require assuming a highly intricate history of introgression losses, as described in lines **381-445**.

While it is possible that a single introgression pulse followed by differential loss of introgressed segments could explain the observed patterns, it would require unusually strong and localized selective pressures acting only on these strains to preserve large heterozygous blocks while allowing erosion and loss in others. Consistent with this, the 7 MA1 strains share a similar number of introgressed genes (from 60 to 96) with other neotropical lineages but have 2x to 6x times more private introgressions, suggesting an additional, lineage-specific event. The proportion of heterozygous private introgressions in these strains is also nearly twice that of the heterozygous

shared introgressions, consistent with a younger hybridization signal. Taken together, these patterns are much more consistent with a secondary introgression event than with differential loss from a single ancestral hybridization.

Still, alternative scenarios suggested by the reviewer cannot be completely ruled out and they are now mentioned in the following lines: Results, **lines 389–392**; Discussion, **lines 508-510**. We do underscore, however, that multiple pulses of introgression are the most parsimonious evolutionary scenario

Comment 2.23. Line 331. Was any attempt made to investigate the pool of introgressed genes with respect to function?

Yes, we performed a GO analysis enrichment of the genes that are present in the introgressed blocks. No significant enrichment was found, at least focusing on those shared among most isolates. We now include a sentence summarizing this analysis in the Results section of the revised manuscript (**lines 372-373**).

Comment 2.24. Line 342. I think it is important to focus on sampling natural environments rather than anthropic ones (see previous comments on how to address the “true” natural history, rather than the recent domestication history).

As described above, we agree about these comments and now provide more carefully worded statements in the revised manuscript (see **lines 532-534**).

Comment 2.25. Line 349. “Finally, the genetic divergence between the MA1 and MA2 clades is associated with the presence of the Sierra Madre Oriental Mountain range, which is found at the edge where the Neotropical and Nearctic biogeographic regions coincide”. Was the actual genetic divergence calculated?

As suggested, the F_{st} values between the Neotropical groups are now reported in **Figure S2**. Notably, the genetic divergence between these clades is higher than that observed between WB3 and Tequila or between MA1 and SAM2, underscoring the significance of this biogeographic barrier in shaping yeast population structure and diversity (see **line 144–145**).

	FG	MA2	MA1	SAM2	Tequila
MA2	0.62				
MA1	0.70	0.43			
SAM2	0.57	0.38	0.39		
Tequila	0.60	0.33	0.32	0.20	
WB3	0.76	0.52	0.54	0.22	0.37

Supplementary Figure S2C

Comment 2.26. Line 397-398. Why not use IQ-TREE with the ‘ASC’ parameter associated with the best model inference? This takes into account that the dataset is SNPs and not coding sequences/proteins.

As suggested, we used IQ-TREE with the ‘ASC’ parameter and replaced all phylogenies in the manuscript with the ones generated using method. We also highlight the consistency of the results regardless of the specific phylogeny-building strategy used (**lines 642-649** and images below).

SACE489_toCONC

Reviewer #3 (Remarks to the Author):

We thank this reviewer for their collaborative review and constructive feedback.

September 5th, 2025

ITEMIZED RESPONSE TO REVIEWERS

Manuscript ID NCOMMS-24-72197-T

Title: "Recurrent introgression and geographical stratification shape *Saccharomyces cerevisiae* in the Neotropics" by Avelar-Rivas et al.

We were encouraged to learn that all reviewers found their original comments thoroughly addressed in our revised manuscript. We also thank the reviewers for their constructive feedback and an outstanding review experience, including attention to the final details in this second round of comments. Below, we provide a point-by-point response to the reviewers' comments.

REVIEWER COMMENTS

Reviewer #1 (Remarks to the Author):

The authors have addressed my previous comments and have improved the manuscript significantly.

*With the re-review of the material, I have a question regarding the models proposed by the authors regarding the formation of the MA1 genotype(s). As pointed out by the authors, the strains within the MA1 clade show far higher variation in the specific introgressed genes than strains within other clades. How do the authors propose that this variation in gene content in the MA1 cluster occurred? Would a full hybridization *cerevisiae* x *paradoxus* event have occurred followed by differential gene loss? Are the various strains still undergoing genome reduction? Could this be a continual process with hybridisation commonly occurring?*

These questions regarding the variation in gene content in the MA1 cluster, are important and stimulating areas we look forward to exploring in future work. The MA1 strains show considerable variation in the introgressed genes that they retain, likely reflecting genome instability and backcrossing. This suggests they are in an ongoing transition from hybridization to stable introgression, though more data and testing are needed to clarify the process. To address this point, we have expanded our discussion starting on line 391 (tracked changes file), with a paragraph that now opens with an explicit reference to the variation within MA1. It suggests possible scenarios, highlights open questions, and mentions potential future work such as experimental and computational tests to estimate the timing of hybridization and backcrossing events, and broader sampling to better capture the evolutionary dynamics of interspecies gene flow.

Reviewer #2 (Remarks to the Author):

I was very pleased to see that the authors carried out a thorough and very careful revision of the manuscript. As far the concerns I raised in my first revision, I see that the authors addressed them, and I do not have are relevant request for change in the manuscript.

There is however one issue, that of the divergence of the Neotropical cluster (Fig. 1B), in which I tend to disagree with the authors. I do not think that further scrutiny will support this divergent placing of the Neotropical cluster that is probably due to a unbalanced sampling of the Neotropical representatives, and perhaps of other groups like the wine clade, by comparison with the representatives of other populations. It is not the number, e.g. phylogenies with more than 1000 or 3000 genomes, but the fine balance of the numbers of genomes of representatives of ALL the populations that is of importance. Please check here <https://doi.org/10.1016/j.cub.2025.04.056> how a probably more balanced sampling (that of course misses the “Mexican Agave 2”) fails to show the neotropical cluster as the utterly divergent lineage seen in Fig. 1. However, I want to stress that I am making this comment “for the record” not as a request for change.

We thank the reviewer for sharing their concern and fully agree on the importance of achieving a more balanced sampling across populations, as fine balance among representatives is indeed critical and may still be partly missing in our dataset. In line with this observation, Fig. 2S of our manuscript already shows that a more balanced sampling reduces the branch length of the Neotropical cluster. A more comprehensive and balanced sampling will be required to clarify the extent and causes of this pattern. To acknowledge this point, we have introduced new text in the results (line 147, tracked changes file) and discussion (line 434, tracked changes file).

*I conclude by congratulating the authors for this very interesting and finely prepared manuscript that I am sure will stand as a relevant contribution for *S. cerevisiae* population biology.*

Minor notes

Line 8, 102, 172, 224, 341, 401, 575 “open agave fermentation”. Why not employing “spontaneous fermentation” which provides the reader with an objective concept, clearer than that of “open fermentation”?

We agree that “spontaneous fermentation” is a clearer and more objective term, and we have incorporated it where appropriate, as suggested. At the same time, we chose to retain “open fermentation” in places where we want to emphasize the physical context, namely, that these fermentations typically occur in open tanks and locations exposed to the natural environment, where microorganisms constantly colonize the agave must.

Line 30 “The Neotropical region, which extendS from...”

Line 97 Agave fermentation strains belong TO a genetically diverse Neotropical cluster

We corrected both errors in the manuscript in line 30 and in the updated line 88 of changes document.

Reviewer #3 (Remarks to the Author):
